# Global energy use and carbon emissions from irrigated agriculture

Jingxiu Qin[1,2], Weili Duan [1] ✉, Shan Zou[1,3], Yaning Chen [1], Wenjing Huang[4] & Lorenzo Rosa [5]

Irrigation is a land management practice with major environmental impacts. However, global energy consumption and carbon emissions resulting from irrigation remain unknown. We assess the worldwide energy consumption and carbon emissions associated with irrigation, while also measuring the potential energy and carbon reductions achievable through the adoption of efficient and low-carbon irrigation practices. Currently, irrigation contributes 216 million metric tons of $CO_2$ emissions and consumes 1896 petajoules of energy annually, representing 15% of greenhouse gas emissions and energy utilized in agricultural operations. Despite only 40% of irrigated agriculture relies on groundwater sources, groundwater pumping accounts for 89% of the total energy consumption in irrigation. Projections indicate that future expansion of irrigation could lead to a 28% increase in energy usage. Embracing highly efficient, low-carbon irrigation methods has the potential to cut energy consumption in half and reduce $CO_2$ emissions by 90%. However, considering country-specific feasibility of mitigation options, global $CO_2$ emissions may only see a 55% reduction. Our research offers comprehensive insights into the energy consumption and carbon emissions associated with irrigation, contributing valuable information that can guide assessments of the viability of irrigation in enhancing adaptive capacity within the agricultural sector.

Seventy percent of worldwide water withdrawals and 80–90% of water consumption are attributed to irrigated agriculture[1]. Irrigation plays a pivotal role in ensuring global food security, contributing to 40% of global food production while utilizing only 22% of the planet's cultivated areas[2,3]. As the challenges of global warming and population growth intensify, exacerbating concerns about water and food security, irrigation emerges as a crucial adaptive measure to address future food crises and the impacts of climate change[4].

Presently, irrigation relies on fossil fuel-based energy for pumping, resulting in the emission of greenhouse gases (GHGs)[5-7]. Numerous studies have quantified GHG emissions within agriculture and food systems[8], covering aspects like land use[9], the production and utilization of synthetic nitrogen fertilizers[10,11] enteric fermentation from livestock production[12], and the entire spectrum of food production, transportation, and consumption[13,14]. Previous studies have provided global and regional datasets detailing GHG emissions related to agriculture[15,16]. Additionally, prior research has assessed indirect GHG emissions linked to irrigation[17], such as methane emissions from reservoirs, ditches, and channels used for irrigation[18], methane and nitrous oxide emissions from rice fields[19], and nitrous oxide emissions under different fertilizer nitrogen use efficiencies[20].

[1]State Key Laboratory of Desert and Oasis Ecology, Key Laboratory of Ecological Safety and Sustainable Development in Arid Lands, Xinjiang Institute of Ecology and Geography, Chinese Academy of Sciences, Urumqi 830011, China. [2]University of Chinese Academy of Sciences, Beijing 100049, China. [3]Akesu National Sation of Observation and Research for Oasis Agro-ecosystem, Akesu, Xinjiang 843017, China. [4]North China University of Water Resources and Electric Power, Zhengzhou 450046, China. [5]Department of Global Ecology, Carnegie Institution for Science, Stanford, CA 94025, USA. ✉e-mail: duanweili@ms.xjb.ac.cn

Furthermore, earlier studies have estimated irrigation energy consumption and GHG emissions specifically from irrigation in China[21], India[22], the Mediterranean region[23], Pakistan[24], and the United States[25]. However, there is a notable gap in studies providing global coverage of energy-related GHG emissions stemming from irrigation. Consequently, the extent to which GHG emissions from irrigation contribute to overall agricultural GHG emissions, and its role in global climate mitigation efforts, remains largely unknown. A comprehensive, globally distributed analysis of energy consumption and GHG emissions inherent to irrigation and pumping systems is imperative for devising effective mitigation strategies toward achieving agriculture with net-zero emissions[8].

While endeavors to diminish GHG emissions have primarily centered around energy and industrial systems[26,27], studies addressing GHG reductions in agriculture, which accounts for 12% of total GHG emissions (7.1 Gt $CO_2$ equivalent per year)[8], have garnered comparatively little attention. Moreover, the emphasis has predominantly been on enhancing the efficiency of irrigation water[28] rather than actively reducing energy consumption and $CO_2$ emissions. Consequently, there is a pressing need for the sustainable development of irrigated agriculture, aiming to enhance food production with reduced reliance on water, energy, and GHG emissions[29].

Here, we quantify global energy consumption and $CO_2$ emissions from irrigation spanning the years 2000–2010, addressing the current gap in understanding farm energy and $CO_2$ emissions. We undertake a spatially explicit analysis of energy and $CO_2$ emissions from both surface and groundwater pumping on a global scale, utilizing a resolution of 10 × 10 km. Our assessment involves quantifying energy use across different irrigation systems—surface, sprinkler, and drip irrigation—as well as pumping systems, encompassing electricity and diesel pumping. The results are then aggregated spatially to quantify country-specific and global energy and $CO_2$ emissions associated with irrigation. Secondly, we explore the phenomenon where groundwater, when pumped, may become supersaturated in carbonate relative to atmospheric pressure, leading to degassing and direct $CO_2$ emissions in irrigated fields[30]. Consequently, we quantify $CO_2$ emissions originating from groundwater degassing and compare these emissions with those related to energy from pumping[30]. Third, our study delves into estimating future energy and $CO_2$ emissions resulting from sustainable irrigation expansion in a 2050 water-efficient and low-carbon scenario. This expansion is contingent on local water availability meeting irrigation water demand under global warming scenarios[4]. Fourth, we evaluate the efficacy of various mitigation interventions and assess their feasibility in achieving a reduction in $CO_2$ emissions within irrigation systems. Finally, we assess the energy and $CO_2$ intensity of irrigation, comparing it with other farm operations such as fertilizers and machinery.

## Results

### Energy and $CO_2$ emissions intensity of irrigation
The energy intensity and $CO_2$ emissions intensity associated with irrigation exhibit significant variations across countries and continents (Fig. 1a, c). Median values for energy intensity and $CO_2$ emissions intensity per unit of irrigation area stand at 2655 MJ/ha and 259 kg $CO_2$/ha, respectively (Fig. 1a, c). Asia registers the highest energy intensity (8.0 GJ/ha) and $CO_2$ emissions intensity per hectare (1063 kg $CO_2$/ha), trailed by Africa (6.7 GJ/ha, 678 kg $CO_2$/ha), South America (6.2 GJ/ha, 506 kg $CO_2$/ha), North America (4.6 GJ/ha, 485 kg $CO_2$/ha), Oceania (4.3 GJ/ha, 375 kg $CO_2$/ha), and Europe (1.9 GJ/ha, 218 kg $CO_2$/ha) (Fig. 1a, c).

Our analysis reveals that the average energy use and $CO_2$ emissions intensity of sprinkler irrigation is the highest (1.8 MJ/m³, 188.4 g $CO_2$/m³), followed by drip irrigation (1.0 MJ/m³, 109.0 g $CO_2$/m³), and surface irrigation (0.5 MJ/m³, 58.5 g $CO_2$/m³) (Fig. 1b, d). Additionally, the energy use and $CO_2$ emissions intensity of diesel pumping (1.2 MJ/

m³, 106.4 g $CO_2$/m³) exceeds that of electric pumping (0.5 MJ/m³, 69.0 g $CO_2$/m³) (Fig. 1b, d).

### Energy consumption and $CO_2$ emissions from irrigation in 2000-2010
Our examination of energy consumption and $CO_2$ emissions from irrigation between 2000 and 2010 reveals significant disparities among countries and continents (Fig. 2 and Supplementary Fig. 4). The global energy consumption attributed to irrigation is 1896 PJ, with Asia and North America accounting for 72% and 14%, respectively (Fig. 2a and Supplementary Fig. 4a). In Asia, India stands out the largest energy consumer with 535 PJ, followed by China (299 PJ), Pakistan (135 PJ), and Iran (121 PJ), constituting 39%, 22%, 10%, and 9% of Asia's total energy consumption from irrigation, respectively (Fig. 2a). In North America, the United States emerges as the foremost energy consumer with 205 PJ, trailed by Mexico with 50 PJ, accounting for 77% and 19% of North America's energy consumption from irrigation, respectively (Fig. 2a). Notably, five countries—India, China, the United States, Pakistan, and Iran—collectively contribute to 68% of the global energy consumption from irrigation (Fig. 2a).

The overall $CO_2$ emissions associated with irrigation stem from both embodied energy consumption and groundwater degassing – the removal of dissolved $CO_2$ from water through degasification (Fig. 2b and Supplementary Figs. 4b and 5a). Our estimation indicates that the global total $CO_2$ emissions from irrigation amount to 222 Mt $CO_2$ per year, with 216 Mt $CO_2$ originating from energy consumption (Fig. 2b) and 6 Mt $CO_2$ from groundwater degassing (Supplementary Fig. 5a). Asia and North America emerge as the primary contributors to $CO_2$ emissions, emitting 164 Mt $CO_2$ and 32 Mt $CO_2$, respectively, collectively constituting 88% of global total $CO_2$ emissions. In contrast, Europe, Africa, Oceania, and South America contribute a combined total of 26 Mt $CO_2$ in irrigation-related $CO_2$ emissions (Fig. 2b). Additionally, the top five countries with the highest $CO_2$ emissions from irrigation due to energy consumption are India, China, the United States, Iran, and Pakistan, emitting 70, 35, 24, 13, and 12 Mt $CO_2$ per year, respectively, accounting for 72% of global $CO_2$ emissions from energy consumption.

When considering $CO_2$ emissions from groundwater degassing, India and the United States stand out the major contributors, emitting 2.9 and 1.4 Mt $CO_2$, respectively (Supplementary Fig. 5a). Notably, in major irrigation-intensive regions of the United States, India, Pakistan, Iran, and Saudi Arabia, $CO_2$ emissions from groundwater degassing account for more than 20% of total $CO_2$ emissions from irrigation (Supplementary Fig. 5b).

### Future energy consumption and $CO_2$ emissions
Sustainable irrigation is irrigation practices that do not deplete groundwater stocks and impair freshwater ecosystems[31,32]. As global warming and food demand increases, sustainable irrigation expansion is an important adaptation solution to future food crises and climate change[3]. The expansion of irrigation inevitably leads to energy consumption and energy-related $CO_2$ emissions. Therefore, we have also conducted an estimation of the energy and $CO_2$ emissions associated with future irrigation expansion. Our assumption is that irrigation will expand in regions where water is expected to be locally available to meet the demand for irrigation water in a climate that is 3 °C warmer—a projected warming level under business-as-usual scenarios[4]. We presuppose that the existing country-specific efficiency of irrigation water usage, encompassing drip, sprinkler, and surface irrigation systems, remains constant in the envisioned scenario of sustainable irrigation expansion in the future. Our estimate indicates that the global additional energy consumption for future irrigation due to sustainable expansion would be 536 PJ, representing 28% of the total irrigation energy consumption in 2000–2010 (Fig. 2a and Supplementary Fig. 6a).

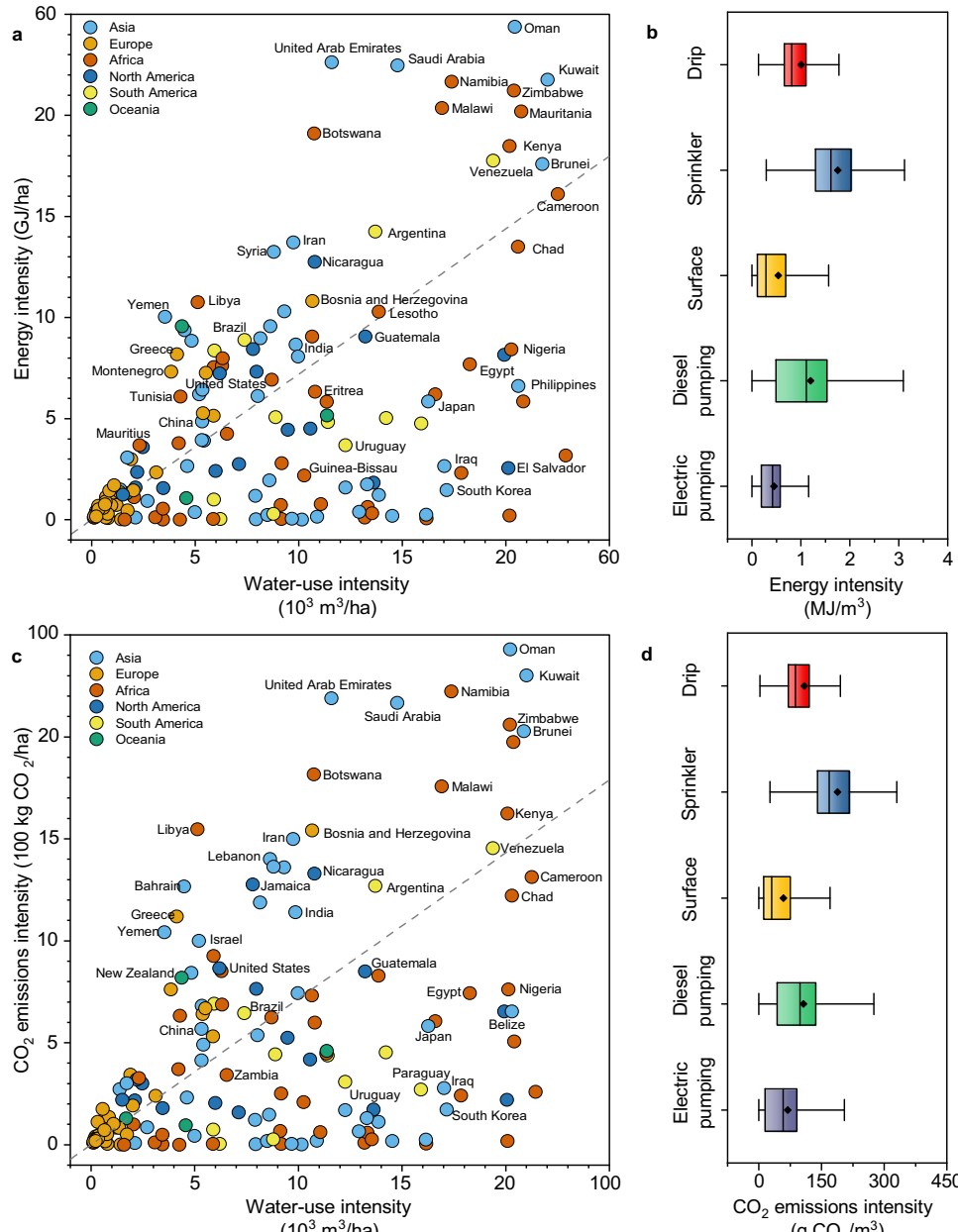

**Fig. 1 | Energy and CO2 emissions intensity as a function of water-use intensity for 159 countries.** The figure shows the comparison of energy and $CO_2$ emissions intensity of different irrigation and pumping systems. **a** Energy intensity is expressed as the ratio of the energy consumed by irrigation to the irrigated area in a country (GJ/ha). **b** The energy intensity of each irrigation system (drip, sprinkler, and surface) or pumping system (diesel pumping and electric pumping) (MJ/m³) represents the energy consumption of five different irrigation and pumping system to pump and deliver one cubic meter of water. **c** $CO_2$ emissions intensity is expressed as the ratio of the carbon dioxide emitted by irrigation to the irrigated area[34] in a country (100 kg $CO_2$/ha). **d** The $CO_2$ emissions intensity of each irrigation system or pumping system is expressed in the same way as the energy intensity (g $CO_2$/m³). Energy and $CO_2$ emissions intensity reflect the average level during the 2000–2010 period. Mean values in the boxplot are shown with diamonds and median values are shown with midlines. Dashed lines in the figure are used to distinguish countries that are above or below the median energy and $CO_2$ emissions intensity per unit of water use.

In North America, Africa, and South America, the energy consumption arising from sustainable irrigation expansion is projected to require an additional 139 PJ, 63 PJ, and 60 PJ, respectively, each exceeding 50% of their current energy consumption (Fig. 2a). Notably, Europe anticipates an additional energy consumption from sustainable irrigation expansion of 148 PJ, which is twice its current energy consumption (Fig. 2a). The United States, India, Russia, Brazil, and Mexico are identified as the top countries with the highest energy consumption from sustainable irrigation expansion, contributing 97 PJ, 49 PJ, 39 PJ, 39 PJ, and 18 PJ, respectively, collectively accounting for 45% of the total energy consumption from sustainable irrigation expansion (Fig. 2a).

Assuming the full adoption of electric pumps and the projected regional carbon intensity of electricity in 2050[33], the additional energy-related $CO_2$ emissions resulting from sustainable irrigation expansion are estimated to be 15 Mt $CO_2$ per year, constituting 7% of the 2000-2010 total energy-related $CO_2$ emissions (Fig. 2b and Supplementary Fig. 6b). India and Russia emerge as the most significant contributors to $CO_2$ emissions from sustainable irrigation expansion, emitting 3 and 2 Mt $CO_2$ per year, respectively.

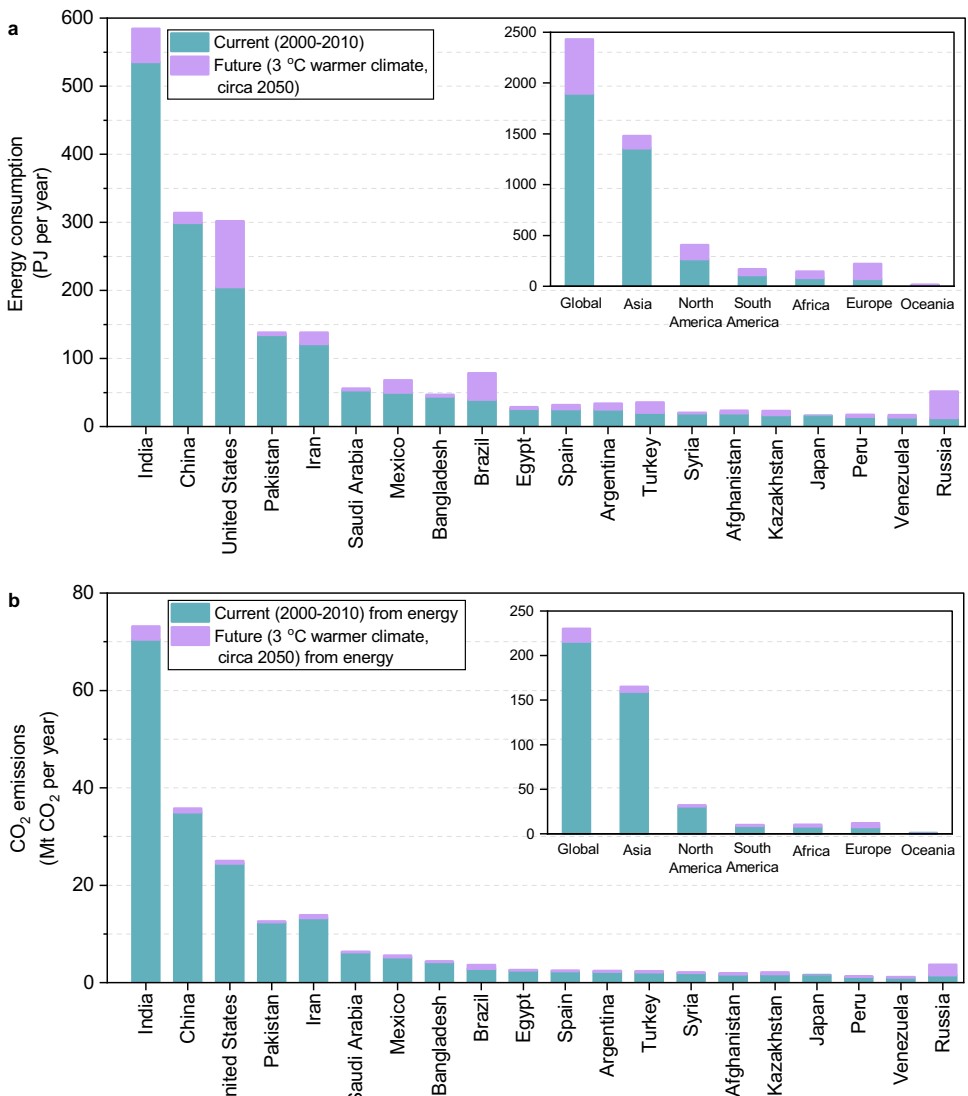

**Fig. 2 | Global energy consumption and CO2 emissions from irrigation.** Country-level energy consumption and $CO_2$ emissions are based on pixel sum statistics. **a** Energy consumption (PJ per year) under 2000–2010 and sustainable irrigation expansion in 3 °C warmer climate. **b** $CO_2$ emissions (Mt $CO_2$ per year) from energy consumption under 2000–2010 and sustainable irrigation expansion of 3 °C warmer climate. We selected the top 20 countries with the highest energy consumption and $CO_2$ emissions. The upper right subgraphs represent a summary of energy consumption and $CO_2$ emissions by regions as well as globally. Geospatial distribution maps are provided in Supplementary Figs. 4 and 6.

## Distribution of energy and $CO_2$ emissions

Figure 3 illustrates the global distribution flow of energy consumption and energy-related $CO_2$ emissions embedded in irrigation and pumping systems, along with irrigation water sources. Despite only 40% of irrigated agriculture being supplied by groundwater[34], energy consumption from groundwater pumping constitutes 89% (1670 PJ per year) of total energy consumption (Fig. 3a). Within this, 74% (1234 PJ per year) of the energy consumption is attributed to diesel pumping, while electric pumping contributes to 26% (436 PJ per year) of the energy usage (Fig. 3a). Energy consumption from surface irrigation systems constitutes 75% (1400 PJ per year) of the overall energy consumption, with energy consumption from sprinkler and drip irrigation accounting for only 21% (388 PJ per year) and 4% (78 PJ per year), respectively. Notably, a significant portion of energy consumption arises from groundwater extraction using diesel pumps combined with surface irrigation system, contributing to 57% (1065 PJ per year) of the total energy consumption.

In terms of $CO_2$ emissions, 90% (193 Mt $CO_2$ per year) are attributed to groundwater pumping, with diesel pumping contributing 57% (110 Mt $CO_2$ per year) and electric pumping contributing 43% (83 Mt $CO_2$ per year) (Fig. 3b). $CO_2$ emissions from surface irrigation systems account for 76% (162 Mt $CO_2$ per year) of the total energy-related $CO_2$ emissions, with 59% (95 Mt $CO_2$ per year) contributed by diesel pumps and 41% (67 Mt $CO_2$ per year) by electric pumps. Conversely, $CO_2$ emissions from sprinkler and drip irrigation constitute only 20% (43 Mt $CO_2$ per year) and 4% (8 Mt $CO_2$ per year), respectively. Additionally, $CO_2$ emissions from groundwater extraction using diesel pumps combined with surface irrigation systems contribute to 45% of the total energy-related $CO_2$ emissions.

## Mitigation options to reduce energy use and $CO_2$ emissions

As the implementation of irrigation and pumping systems directly contributes to energy consumption and, consequently, energy-related $CO_2$ emissions, mitigation measures must address both the efficiency of irrigation equipment and the carbon intensity of energy. To explore viable options for reducing energy and $CO_2$ emissions, we examine two main scenarios, namely enhancing irrigation systems to reduce water-

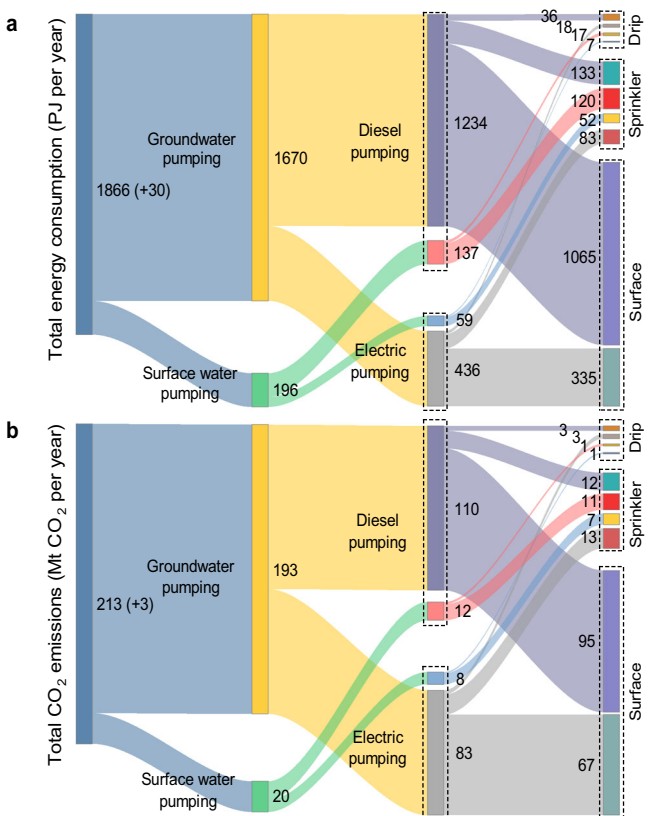

**Fig. 3 | Sankey diagram of the distribution of energy consumption and energy-related CO2 emissions from irrigation. a** Energy consumption (PJ per year) during 2000–2010. **b** CO$_2$ emissions from embodied energy consumption (Mt CO$_2$ per year). In the figure, the total energy consumption and CO$_2$ emissions do not include those (values in parentheses) from natural gas pumping. Geospatial distribution maps are provided in Supplementary Figs. 7-13.

use intensity and transitioning pumping systems to electric pumping while utilizing low-carbon electricity (see Methods).

Figure 4 shows global energy and CO$_2$ emissions under these distinct scenarios. In comparison to the 2000-2010 period, both energy consumption and energy-related CO$_2$ emissions from irrigation are cut in half under the drip irrigation scenario (Fig. 4). However, the sprinkler scenario results in an increase of 39% (743 PJ per year) in energy consumption and CO$_2$ emissions (Fig. 4a). Shifting all diesel pumping to electric pumping leads to a 51% (966 PJ per year) reduction in energy consumption (Fig. 4a). Furthermore, under the electric pumping scenario, where the electricity sources include 2000–2010 electricity mix, solar, wind, nuclear, hydropower, and projected electricity mix in 2050, the energy-related CO$_2$ emissions from irrigation are reduced to 175, 11, 3, 3, 6, and 6 Mt CO$_2$ per year, respectively (Fig. 4b). Notably, the potential for CO$_2$ emissions mitigation exceeds 90% under low-carbon electricity scenarios and is limited to 19% under the 2000–2010 electricity mix scenario (Fig. 4b).

**Feasibility of solutions to reduce energy use and CO$_2$ emissions**
Through our examination of solutions aimed at reducing energy and CO$_2$ emissions, we demonstrate that both energy consumption and CO$_2$ emissions can be significantly diminished under drip and electric pumping scenarios. Notably, in the electric pumping scenario, a substantial reduction in CO$_2$ emissions is achievable only when electricity is low-carbon (Fig. 4). As a result, the proportion of low-carbon electricity in 2050 will play a pivotal role in determining the extent to which CO$_2$ emissions from irrigation can be reduced. The feasibility of adopting drip irrigation becomes a crucial factor in selecting strategies

for energy and CO$_2$ emissions reduction. For instance, given that drip irrigation is not applicable to rice cultivation, and not all countries will adopt low-carbon electricity by 2050[35], significant reductions in CO$_2$ emissions may not be attainable. Consequently, this section focuses on analyzing the feasibility of solutions aiming at reducing CO$_2$ emissions.

Figure 5 shows the feasibility of drip irrigation and low-carbon electricity, as well as the potential contribution to reducing energy-related CO$_2$ emissions for each country. The feasibility of low-carbon electricity is greater than drip irrigation. The Middle East and North Africa and Western Europe have higher feasibility of drip irrigation, with 40% and 29%, respectively (Fig. 5a). The Middle East and North Africa, Southeast Asia and Oceania, Western Europe, Sub-Saharan Africa, East Asia, and Latin America and the Caribbean have higher feasibility of low-carbon electricity, both exceeding 30% (Fig. 5a).

The feasibility of low-carbon electricity, that is, how much the share of low-carbon electricity can increase by 2050 compared with 2000-2010, determines how much CO$_2$ emissions can be reduced by 2050. Figure 5b shows that 55% of global energy-related CO$_2$ emissions are reduced through a combination of low-carbon electricity and drip, with 82% of the reduction being contributed by low-carbon electricity and 18% by drip irrigation. Middle East and North Africa, Northern America, and Western Europe can achieve over 60% reduction in energy-related CO$_2$ emissions, with drip irrigation contributing 30% of the reduction in the Middle East and North Africa (Fig. 5b). In South Asian and East Asia with intensive irrigation, over 50% of energy-related CO$_2$ emissions are reduced through a combination of low-carbon electricity and drip (Fig. 5b). However, in Eastern Europe and Central Asia, only 15% of energy-related CO$_2$ emissions are reduced through a combination of low-carbon electricity and drip (Fig. 5b).

**Irrigation contribution to on farm energy use**
Based on the analysis of energy intensity and CO$_2$ emissions intensity of irrigation at the national scale (Fig. 1a,c), we further analyze irrigation contribution to farm energy use. Figure 6 shows the comparison of energy input and carbon emissions intensity of irrigation with total energy input and carbon emissions intensity on farm in sub-regions worldwide. Globally, energy input intensity of irrigation accounts for 32% of global energy input intensity, and over 50% in the sub-Saharan Africa, Middle East and North Africa, South Asia, and Latin America and the Caribbean (Fig. 6a). Accordingly, CO$_2$ emissions intensity of irrigation accounts for 33% of global CO$_2$ emissions intensity, and over 50% in sub-Saharan Africa, Middle East and North Africa, and South Asia (Fig. 6b). In addition, the largest energy input intensity and CO$_2$ emission intensity of North America also come from irrigation.

Based on the cropland area from FAOSTAT[15], we estimate global energy consumption (PJ) and the corresponding carbon emissions (Mt CO$_2$e) from fertilizers, machinery, and fuel (Supplementary Table 8). We find that energy consumption and carbon emissions from irrigation account for approximately 15% of the total energy consumption and carbon emissions in agriculture (Supplementary Table 8).

## Discussion
Since the Green Revolution of the 1960s, global crop production has increased by 3.7 times[15] due to the intensification, mechanization, and modernization of agricultural systems[36]. However, gains in yield have come at a considerable cost in terms of increased energy input and considerable environmental footprint[37,38]. Irrigation development, as the concentrated embodiment of agricultural intensification, involved high energy consumption, led to reliance on fossil fuel, and CO$_2$ emissions (Fig. 2). In addition, energy input and carbon emissions from irrigation contribute significantly to total energy consumption and carbon emissions in agriculture (Fig. 6 and Supplementary Table 8).

High energy input and carbon footprint of irrigation in turn would potentially threaten the growth and stability of food production worldwide, especially in regions heavily dependent on fossil fuels[37].

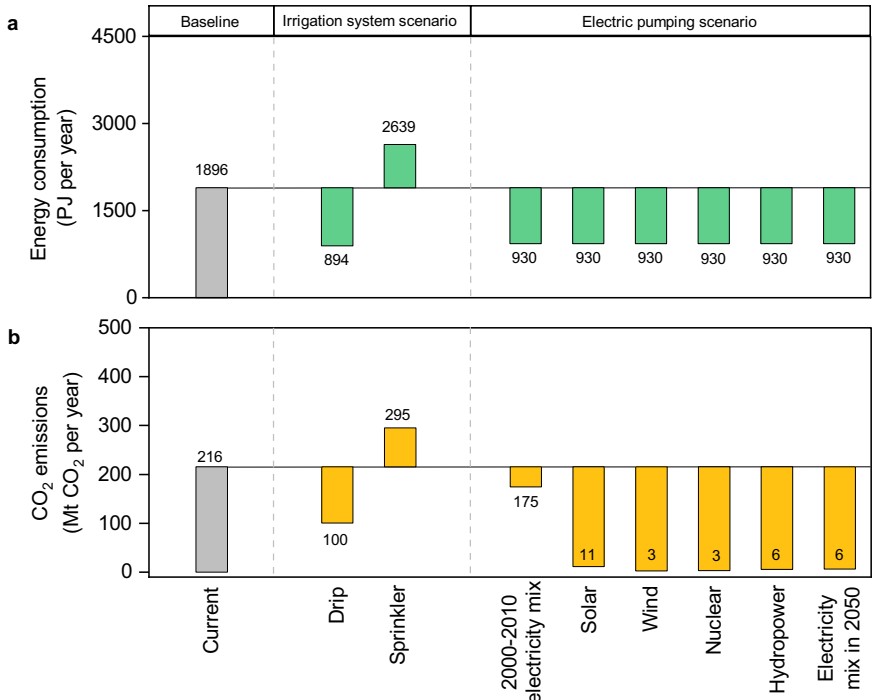

**Fig. 4 | Mitigation potentials of energy use and CO2 emissions under different scenarios. a** The sprinkler scenario and drip scenario have different water-use intensities. For the electric pumping scenario, the energy consumption of irrigation does not change with the $CO_2$ intensity of electricity. **b** In sprinkler and drip scenarios, $CO_2$ intensity of energy (electricity during 2000–2010) remains unchanged. In electric pumping scenarios, electricity comes from 2000-2010 electricity mix, solar, wind, nuclear, hydropower, or a mix of the four low-carbon electricity according to IEA net-zero by 2050 projections[35]. The baseline scenario reflects energy consumption and energy-related $CO_2$ emissions during 2000–2010. The 2000-2010 electricity mix represents the electricity production structure in 2000-2010, where the electricity comes from fossil fuels, nuclear, hydro, geothermal, solar, wind, tide, wave, ocean and biofuels[55]. Geospatial distribution maps are provided in Supplementary Figs. 14-16.

Furthermore, the high energy input of irrigation also increases the pressure on the energy supply system and competition for energy from other sectors. In Pakistan and Bangladesh, the energy consumption of irrigation alone accounts for 4% of the total energy supply in 2000-2010 (Supplementary Fig. 19a). With future sustainable irrigation expansion (Fig. 2), the additional energy consumption of irrigation will add pressure on the energy supply in African and European countries (Supplementary Fig. 19b).

We provide solutions for achieving low energy consumption and carbon emissions as well as highly efficient water use in irrigated agriculture (Figs. 4 and 5). Drip and sprinkler represent two water-efficient irrigation systems. Still, our results show that sprinkler irrigation system has higher energy and $CO_2$ emissions intensity than surface irrigation and does not reduce energy use and $CO_2$ emissions of irrigation globally (Figs. 1b, d and 4). Therefore, priority should be given to drip irrigation system in the deployment of farm infrastructure. The exception is when switching from gravity irrigation to drip irrigation increases energy consumption and $CO_2$ emissions of irrigation, where there is a trade-off between water savings and reductions of energy and $CO_2$ emissions[39], as exemplified by Sudan and Ethiopia (Supplementary Figs. 4 and 14). In this case, a benefit-cost analysis should be incorporated into the trade-off. If carbon emissions are also considered as an investment cost, then the cost increase induced by the adoption of drip irrigation systems includes the initial capital investment of equipment with benefits throughout the life cycle, usually 15–25 years, the converted cost of energy input, and carbon taxes[38]. The economic returns from investments in drip irrigation technology include improvement in production, a shift in cropping rotation, and water and fertilizer savings[38,40]. If the benefits are greater than the costs, the investment in drip irrigation systems can be treated as economically viable in countries like Sudan and Ethiopia.

The benefit-cost analysis also applies to other countries around the world.

Remarkably, drip irrigation is not applicable to all crops (Supplementary Table 6), and its contribution to reducing global $CO_2$ emissions of irrigation is limited (Figs. 4b and 5b). Another solution in our study is to switch from energy-intensive diesel to efficient electric pumping (Fig. 1b) and use low-carbon electricity. Low-carbon electric pumps have a significant effect on reducing $CO_2$ emissions of irrigation (Figs. 4b and 5b), and have the same efficacy as drip irrigation in reducing energy consumption of irrigation (Fig. 4a). Low-carbon electricity brings substantial long-term benefits to the country, such as reducing reliance on fossil fuels[41]. However, from farmer's perspective, the reduction of $CO_2$ emissions is not a priority. An introduced carbon price may facilitate the adoption of low-carbon electricity, but it may also erode the relative profitability of irrigated crop production[40,42]. Therefore, efforts should be put into the development of low-carbon electricity and cost reduction for countries worldwide in the future. On the other hand, global energy consumption and $CO_2$ emissions from irrigation are dominated by groundwater pumping (Fig. 3). In this case, it is recommended to give priority to the use of surface water and shallow groundwater for irrigation. Meanwhile, the management of groundwater resources should be strengthened to prevent the decline in groundwater levels from offsetting the energy efficiency gains from the adoption of drip irrigation systems[43,44].

Currently, spatially explicit estimates of global irrigation water withdrawals for irrigation are still before 2010[45], which prevents our study from providing the most time-sensitive analysis of global energy consumption and $CO_2$ emissions of irrigation. However, previous studies have shown that irrigation water withdrawal is driven by irrigated area[46]. Thus, we still can provide an updated understanding of global energy consumption and $CO_2$ emissions of irrigation after 2010,

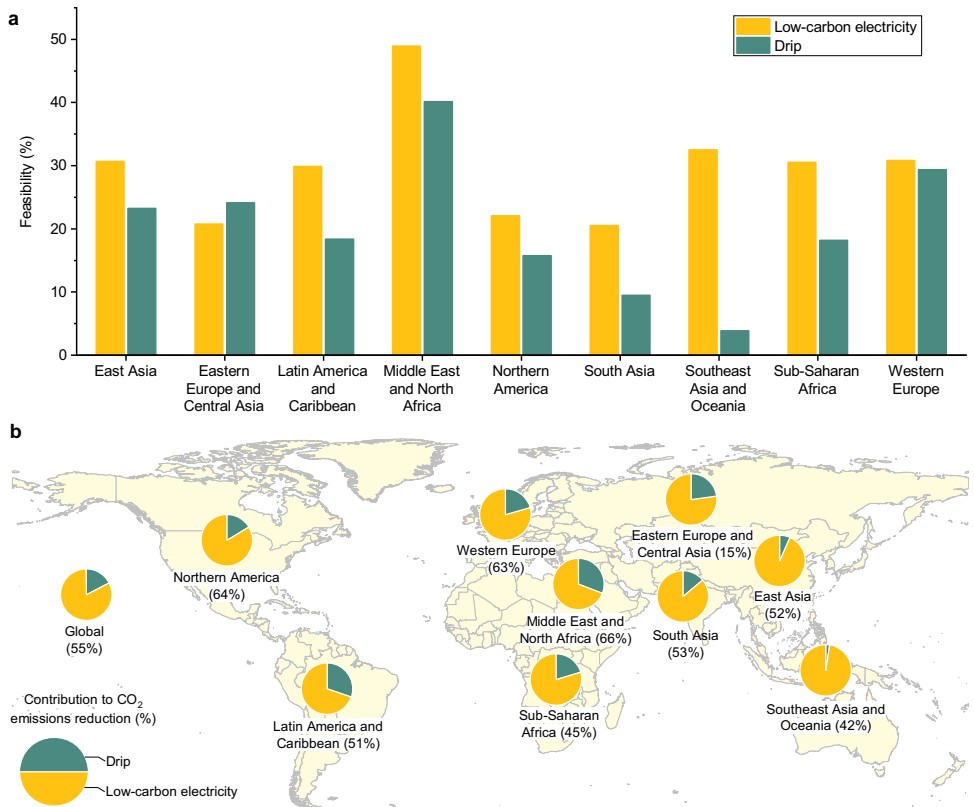

**Fig. 5 | Feasibility of solutions to reduce CO2 emissions of irrigation on a country-level scale. a** Feasibility (%) of drip and low-carbon electricity. **b** Potential contribution (%) of drip and low-carbon electricity to energy-related $CO_2$ emissions reduction based on the feasibility analysis. The pie chart shows the contribution ratio of low-carbon electricity and drip to energy-related $CO_2$ emissions reduction. The values at the bottom of the pie chart represent the total contribution due to a combination of the two solutions. For a more detailed comparison of regional differences, we further divided the six continents into nine sub-regions (See Source Data for the rationale of the classification). Country-level feasibility of drip and low-carbon electricity is provided in Supplementary Fig. 18. GS (2016) 1966.

assuming all other conditions except irrigated areas remain as in 2000-2010. Based on energy and $CO_2$ emissions intensity per unit of irrigated area (Fig. 1 a,c) and country-specific irrigated area derived from FAO AQUASTAT[47] in 2020, global energy consumption and $CO_2$ emissions of irrigation increased by about 14% in 2020 compared to 2000–2010, with significant increases in some African countries (Supplementary Fig. 23).

Our study sheds light on the previously uncharted territory of global energy consumption and carbon emissions associated with irrigation. This research not only provides a comprehensive understanding of global direct energy use and associated $CO_2$ emissions from irrigation but also charts a path forward aiming for less water, energy, and $CO_2$ emissions in irrigated agriculture. Furthermore, previous work showed biophysical limits to irrigation showing where water is locally available to meet crop water demand[3]. However, irrigation is not only influenced by water availability but also socio-economic factors, a phenomenon known as agricultural economic water scarcity[48]. Mapping at high resolution the energy and carbon emissions from irrigation allows us to understand where energy will be a barrier to irrigation. Our study provides detailed information on energy usage from irrigation and can inform on the feasibility of irrigation to increase adaptive capacity in the agricultural sector.

## Methods

### Energy consumption and energy-related $CO_2$ emissions estimates

Irrigation energy use is a function of the volume of irrigation water and of the total pressure head[7] crucially affected by irrigation systems (drip, sprinkler, and surface irrigation), pumping systems (diesel

pumping and electric pumping), and irrigation water source (surface or groundwater) (Supplementary Table 1). Global groundwater table depth datasets are derived from Fan et al.[49]. The percentage of surface water and groundwater for irrigation is taken from Siebert et al.[50]. Country-level datasets on the proportion of drip, sprinkler, and surface irrigation are obtained from Jägermeyr et al.[28]. Irrigation water withdrawal datasets reconstructed based on the global hydrological model LPJmL during 2000–2010 are derived from Huang et al.[51], which are calibrated and validated by using reported data from FAO AQUASTAT[47] and USGS[52]. The groundwater table depth and irrigation water withdrawal datasets were resampled from 30 arc-second and 30 arc-minutes to a 5 arc-minutes resolution using the nearest neighbor method to spatially match the datasets on the proportion of irrigation water source.

We assume that typical operating pressures for surface, drip, and sprinkler irrigation are 0.41 bar[53] (this is set to 0 for surface water sources), 1 bar and 3 bar[23], respectively. The operating friction losses of the piped distribution system are equal to 0.69 bar for all systems derived from Brown et al.[54]. Specifically, the energy requirement can be calculated using Eqs. (1) and (2):

$$EQ = \frac{V \times TH}{367 \times \eta} \tag{1}$$

$$TH = Lift + D + H + f_{losses} \tag{2}$$

where $EQ$ (kW h) is energy requirement; $V$ (m$^3$) is irrigation water volume; $TH$ (m) is the total water pressure head calculated as the sum of the lift from the water table to the ground surface, drawdown depth

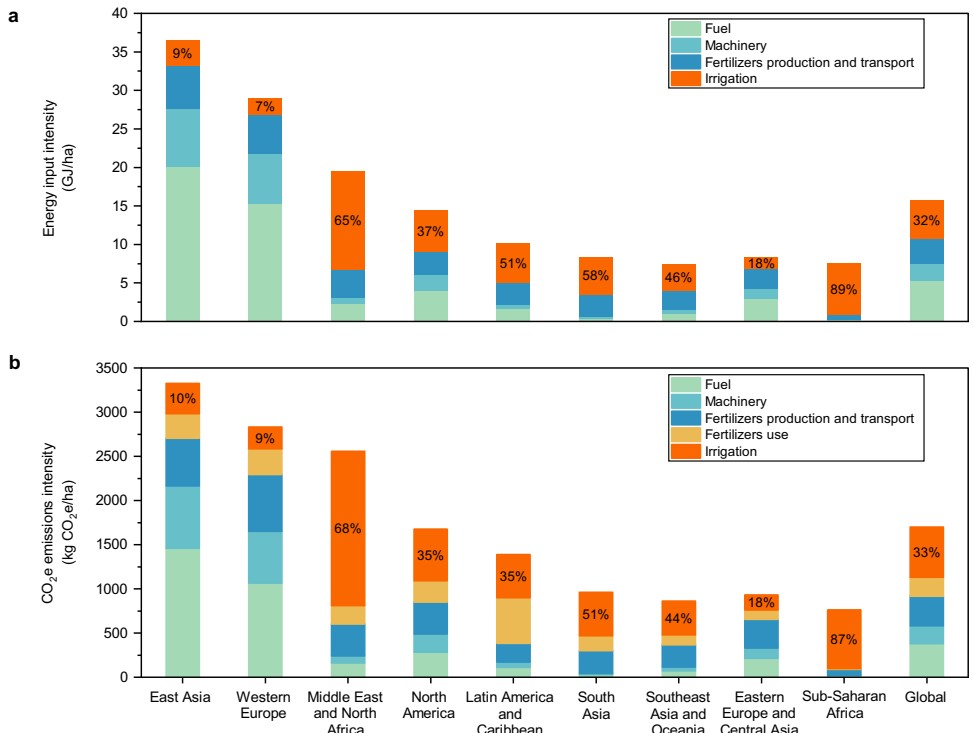

**Fig. 6 | Contribution of irrigation energy input intensity to total energy input intensity and comparison of CO2 emissions intensity generated by energy inputs intensity during 2000–2010. a** Energy inputs intensity comes from irrigation, fertilizers (N, $P_2O_5$, and $K_2O$) production and transport, machinery (includes tractors, harvesters, and threshers[15]) manufacturing, and fuel consumption of machines. **b** Besides GHG emissions generated by the four energy inputs, soil emissions from N fertilizers use are considered. The useful life of machinery is assumed to be a 20-year average. The fuel used in the machinery mainly considers liquefied petroleum gas, motor gasoline, and gas-diesel oils. Other greenhouse gas emissions (e.g. $CH_4$ and $N_2O$) generated by energy inputs are converted to carbon dioxide equivalents ($CO_2$e), using global warming potential (GWP) values of the IPCC AR5 with no climate feedback (GWP-$CH_4$ = 28; GWP-$N_2O$ = 265)[68].

(D, both from cone of depression and additional drawdown from well efficiency), operating pressure (H) and friction losses of pipe ($f_{losses}$); $\eta$ represent the efficiency of the pump and prime mover (%). The prime mover efficiency depends on energy source (mainly diesel and electricity). The lift parameter is represented by groundwater table depth. Due to the lack of global country-level information on the proportion of irrigation pumps (diesel and electric), we estimated the ratio of electric to diesel pumps by indirectly estimating grid coverage in irrigated areas at the pixel scale. According to the best statistical data, county-scale irrigation pump information (diesel, electricity, and natural gas pumps) was used separately for the United States (Supplementary Methods Section 1.4). Estimations of drawdown depth, the efficiency of the pump and prime mover, and proportionality for irrigation pumps are described in detail in Supplementary Methods Section 1.1, 1.2, and 1.3.

The energy-related $CO_2$ emission of irrigation can be calculated as the energy requirement of irrigation multiplied by the carbon emission factor. The energy-related $CO_2$ emission from irrigation can be calculated using Eq. (3):

$$C_e = E_c \times C_{ef} \qquad (3)$$

where $C_e$ (g $CO_2$) is $CO_2$ emissions; $E_c$ (kWh) is energy consumption; $C_{ef}$ (g $CO_2$/kWh) is $CO_2$ emission factor of energy consumption. The national-scale carbon intensity of electricity datasets during 2000–2010 are derived from IEA[55] and Our World in Data[56], which depends on the source of electricity. Because the carbon emission intensity of electricity is influenced by the electricity trade, we considered the influence of the electricity trade (Supplementary Method Section 1.5). $CO_2$ emissions from diesel to produce 1 kWh of

energy are equivalent to 320.21 g $CO_2$[57]. Supplementary Discussion Section 2.1 provides an analysis of the precision of our results.

## $CO_2$ emissions from groundwater degassing

Groundwater water is generally supersaturated in $CO_2$ compared to the overlying atmosphere, and this water-air gradient leads to $CO_2$ degassing when groundwater is pumped to the surface. Groundwater degassing is a source of $CO_2$ emissions from irrigation[30]. However, $CO_2$ emissions from groundwater degassing caused by irrigation were unquantified. The $CO_2$ emissions from groundwater degassing caused by irrigation can be calculated using Eq. (4):

$$C_{GD} = V_{GD} \times R_{IWW} \times C_{GW} \qquad (4)$$

where $C_{GD}$ is $CO_2$ emissions from groundwater degassing caused by irrigation; $V_{GD}$ is groundwater volume due to irrigation ($m^3$); $R_{IWW}$ is the ratio of groundwater withdrawal for irrigated to total groundwater withdrawal; $C_{GW}$ is $CO_2$ concentration in groundwater. We used global average annual groundwater pumping datasets during 2000–2009 with a resolution of 0.5° derived from Döll et al.[58]. The datasets are based on the WaterGAP 2.2 model combined with local well observation or GRACE satellite observation, which can largely reduce uncertainty inherent in flux-based method[59]. Groundwater is mainly used for irrigation, domestic and manufacturing, and the proportion of groundwater use in the domestic and manufacturing sectors suggested by Döll et al.[58] is 36% and 26%, respectively. Reconstructed sectoral water withdrawals datasets were obtained from Huang et al.[51]. A survey of water quality for groundwater aquifer of the United States shows that the 25% and 75% quantiles of bicarbonate concentration are 95 mg/L and 293 mg/L, respectively[60]. We assume that the global bicarbonate concentration of groundwater is likely like that measured

in the United States. This is subsequently converted to $CO_2$ concentration according to Eq. (5):

$$CO_2 Concentration = \frac{1}{2} HCO_3^- \times \frac{44}{61} \tag{5}$$

Finally, the $CO_2$ concentration of groundwater ranges from 34.26 mg/L to 105.67 mg/L.

## Energy and $CO_2$ emissions intensity

Based on the estimation of energy and $CO_2$ emissions, we compared the differences in energy use intensity (GJ/ha) and $CO_2$ emissions intensity (kg $CO_2$/ha) between countries. Using geospatial information from irrigation systems and pumping systems, we compared spatially explicit differences in energy and $CO_2$ emissions per unit of water use between different irrigation and pumping systems.

The energy consumption for irrigation comes directly from the pumping and delivery of water, with diesel and electricity consumption and indirect emissions of carbon dioxide. However, the energy and $CO_2$ emissions intensity of the same irrigation and pumping system is different due to the difference in irrigation water source (surface water and groundwater) and groundwater table depth. Furthermore, differences in operating pressure and pumping efficiency (Supplementary Methods Section 1.2) make the average energy and $CO_2$ emissions intensity different between irrigation systems and pumping systems.

## Setting scenarios for mitigation options

To test the impact of mitigation options, the 2000–2010 condition is defined as a baseline. Furthermore, we selected two main options for reducing energy and $CO_2$ emissions and established multiple scenarios under each option.

The first scenario is to upgrade the agricultural irrigation systems. Different irrigation systems reflect different irrigation efficiency, implying different water-saving efficiency. Globally, drip irrigation will save 43% and 68% of irrigation water withdrawal compared with sprinkler and surface irrigation, respectively, whereas sprinklers will save 44% of irrigation water withdrawal compared to surface irrigation as proposed by Jägermeyr et al.[28] (Supplementary Table 4). Therefore, based on the current irrigation system, two more efficient irrigation scenarios are represented by upgrading surface irrigation to sprinklers and to drip irrigation. We defined the first scenario as a sprinkler scenario and the second scenario as a drip scenario (Supplementary Table 5). Furthermore, we tested the effects of drip and sprinkler irrigation efficiency on energy consumption and $CO_2$ emissions by changing water-saving efficiency by 5% (Supplementary Method Section 1.10 and Supplementary Table 7).

The other option is to reduce $CO_2$ emissions by low-carbon electricity, which comes from solar, wind, nuclear, hydropower, and a mix of the four low-carbon electricity sources by 2050. However, the abstraction and application of irrigation water do not depend solely on electric pumping; diesel pumping also accounts for a considerable proportion in irrigated agriculture (Supplementary Table 3). Therefore, we converted all diesel pumping to electric pumping (electric pumping scenario). The carbon footprint of solar, wind, nuclear, and hydropower based on lifecycle assessment is 44, 11, 12, and 23 g $CO_2$/kWh, respectively[61,62]. The carbon footprint (25 g $CO_2$/kWh) of electricity mix is estimated based on the carbon footprint as well as electricity generation by 2050 of these four powers (Supplementary Method Section 1.6). Likewise, we evaluated the uncertainty of carbon footprint for solar, wind, nuclear, and hydropower on mitigation potential (Supplementary Method Section 1.10 and Supplementary Table 7).

## Feasibility of mitigation options

Based on the scenario setting for reducing energy and $CO_2$ emissions, we further analyzed the feasibility of mitigation options. First, we evaluated the feasibility of drip irrigation and low-carbon electricity, as well as the potential contribution of drip and low-carbon electricity. The feasibility and potential contribution of drip irrigation and low-carbon electricity can be calculated using Eqs. (6) – (9):

$$F_{Drip} = \frac{IWC_{Drip}}{IWC_{Total}} - Drip_{Cur} \tag{6}$$

$$F_{Low} = Low_{2050} - Drip_{Cur} \tag{7}$$

$$C_{Drip} = \frac{F_{Drip}}{1 - Drip_{Cur}} \times C_{DripS} \tag{8}$$

$$C_{Low} = \frac{F_{Low}}{1 - Low_{Cur}} \times C_{LowS} \tag{9}$$

where $F_{Drip}$ is the feasibility of drip irrigation; $IWC_{Drip}$ is the irrigation water consumption of crops that can adopt drip irrigation (Supplementary Table 6); $IWC_{Total}$ is the total irrigation water consumption for 26 crops; $Drip_{Cur}$ is the current ratio of drip. In this study, the irrigation water consumption of 26 crops from 2000–2010 is calculated based on the WATNEEDS model[63] (Supplementary Method Section 1.7). The $F_{Low}$ is the feasibility of low-carbon electricity; $Low_{2050}$ is the share of low-carbon electricity by 2050[33]; $Low_{cur}$ is the share of low-carbon electricity during 2000–2010. $C_{Drip}$ represents the potential contribution of drip; $C_{DripS}$ represents the contribution of $CO_2$ emissions reduction under the drip scenario. $C_{Low}$ represents the potential contribution of low-carbon electricity; $C_{LowS}$ represents the average contribution under the electric pumping scenario where electricity comes from wind, solar, nuclear, and hydropower. We used regional (Europe, Asia, Africa, Middle-East, North America, Latin America, Commonwealth of Independent States (CIS), and Pacific) low-carbon electricity targets for 2050, under which current NDC's (Nationally Determined Contributions) emission targets for 2030 can successfully achieve, as well as a continuation of consistent efforts post-2030.

For a combination of low-carbon electricity and drip irrigation, the contribution value of $CO_2$ emissions reduction can be calculated using Eq. (10):

$$1 - (1 - C_{Drip}) \times (1 - C_{Low}) \tag{10}$$

The feasibility of low-carbon electricity is set to 0, when the 2000-2010 national share of low-carbon electricity exceeds the projected regional targets by 2050 (Supplementary Discussion Section 2.3).

## Other energy inputs and $CO_2$ emissions

From the FAOSTAT database[15], we obtained fertilizers (N, $P_2O_5$, and $K_2O$) use (kg/ha) and cropland area data for each country during 2000–2010. Cropland areas were used to calculate inputs per hectare. We also obtained machinery (number of tractors, harvesters and threshers) data from the FAOSTAT database during 2000–2005[15]. Furthermore, we converted the physical quantities to energy units (GJ/ha) by using time-varying energy conversion factors, which can be found in Pellegrini et al.[36]. For machinery, we assumed an average lifespan of 20 years with an energy conversion factor of 8.35 GJ/t based on Pellegrini et al.[36].

We further converted energy input intensity (GJ/ha) to $CO_2$ emissions intensity (kg $CO_2$e/ha) by using carbon emissions factors from the literature. For the production and transport of N, $P_2O_5$, and $K_2O$ fertilizers, we used the regional (Western Europe, Eastern Europe, Central and South America, Asia, Australia, New Zealand, and global

average for the rest of the countries) emission factors (kg $CO_2$e/kg)[64]. Furthermore, we used the regional (Africa, East Asia, Europe, Latin America, North America, Oceania, South Asia, CIS, and global average for the rest of the countries) emission factors for N fertilizers use (direct soil emissions)[65]. For machinery, we used an average emission factor of 95 kg $CO_2$/GJ for the three machineries[66,67] (Supplementary Method Section 1.9). Due to the lack of information on the types and corresponding proportions of fuel consumed by machinery, we assumed that the fuel consumed by these three kinds of machinery is mainly derived from liquefied petroleum gas, motor gasoline, and gas-diesel oils, which emit $CO_2$, $CH_4$, and $N_2O$, and used average emission factor for the three fuels from the IPCC[68]. Then, we used the total cropland area from FAOSTAT[15] to estimate total energy inputs and GHG emissions from fertilizers, machinery and fuel during 2000–2010. Due to the lack of reliable data for pesticides[69], our analysis did not consider the energy input of pesticides.

### Caveats

The proportion of surface water and groundwater plays a crucial role in accurately estimating global energy consumption and $CO_2$ emission for irrigation. In this study, the global average annual irrigation water withdrawals were 2588 km³ during 2000–2010, which agrees with the results (2673 km³ in 2012) reported by FAO's AQUASTAT[47]. However, because the irrigation water sources proportion data provided by Siebert et al[50]. and irrigation water withdrawal data provided by Huang et al[51]. do not completely match spatially; only 2451 km³ of water are involved in the estimation of energy and $CO_2$ emissions. Therefore, the results in our study are a lower-bound estimate of $CO_2$ emissions from irrigation.

Due to the lack of country-level information on irrigation pumps, we used the proportion of the total irrigated area covered by the global grid network as the proportion of the electric pump, and the rest of the irrigated areas that are not connected to the grid used diesel pumps. Although our estimation results are relatively consistent with the results of previous literature surveys in eight countries (Supplementary Fig. 21 and Supplementary Table 3), there are still uncertainties.

When we calculated the mitigation potential of low-carbon electricity to $CO_2$ emissions, typical values of carbon footprints of solar, wind, nuclear, and hydropower were assumed. However, differences in technology levels between countries and wide ranges of carbon footprints for low-carbon electricity throughout the life cycle led to uncertainties in the mitigation potential of low-carbon electricity on $CO_2$ emissions of irrigation (Supplementary Table 7). Considering the availability of data, we use regional or fixed conversion factors when estimating energy inputs and corresponding carbon emissions of other farm operations, such as emission factors for the use of liquefied petroleum gas, motor gasoline, and gas-diesel oils, which indirectly affects the proportion of irrigation energy input in the total energy input on farms.

In light of the ambitious renewable energy policies reported at COP28 (28th Conference of the Parties to the United Nations Framework Convention on Climate Change) aimed at achieving net-zero emissions targets by 2050[70], our findings may underestimate the potential for carbon emissions reduction from irrigation by 2050. Notably, major consumer of irrigation energy—India, China, Pakistan, and Iran—have not committed to tripling nuclear energy nor increasing renewable power generation capacity threefold by 2030. Consequently, while global carbon emissions from the energy sector are projected to significantly decrease by 2050, reducing carbon emissions from irrigation will necessitate these countries to make greater commitments towards renewable energy adoption. Furthermore, the renewable energy policies outlined at COP28 do not delineate the proportion of low-carbon electricity expected by 2050 for individual countries or continents, hindering precise estimations of the potential

reduction in irrigation carbon emissions. Our study offers a framework for evaluating the potential reduction in irrigation-related carbon emissions, enabling updates to our findings as new data becomes available.

### Reporting summary

Further information on research design is available in the Nature Portfolio Reporting Summary linked to this article.

## Data availability

This work used data collected from a variety of literatures and publicly available sources, which are be listed in the main text and Supplementary material. All analyses are based on these collected datasets. Results from all analyses are available in Source data as Excel spreadsheets alongside the paper. Data for the main results of this study are publicly available from https://doi.org/10.5281/zenodo.10118986. Source data are provided with this paper.

## Code availability

The data were analyzed with the statistical software MATLAB and Origin. The script is publicly available from https://doi.org/10.6084/m9.figshare.25392874.v1.

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

## Acknowledgements
W.D. was supported by the National Natural Science Foundation of China (grant no. 42488201, 42122004), the West Light Foundation of the Chinese Academy of Sciences (grant no. xbzg-zdsys-202208), and the Tianshan Talent Training Program (grant no. 2023TSYCLJ0050). We thank Stefan Sibert for providing suggestions on the draft manuscript.

## Author contributions
J.Q. and W.D. contributed to conceptualization, methodology, writing - review & editing; W.D. and Y.C. contributed to supervision, funding acquisition, writing-review & editing; S.Z. contributed to supervision, writing - review & editing; W.H. contributed to supervision, writing-review & editing; L.R. contributed to conceptualization, methodology, supervision, writing-review & editing. All authors approved the final version of this manuscript.

## Competing interests
The authors declare no competing interests.
