## [Peer Review File · Nature Communications]

Global energy use and carbon emissions from irrigated agricultureReviewers' Comments:

Reviewer #2:

Remarks to the Author:

This is a very interesting paper that analyses the irrigation related energy consumption and carbon emissions globally. Additionally, the paper quantifies the energy and carbon savings that can be achieved through adoption of low-carbon irrigation interventions. This paper carries significant importance in the context when the whole world is directed towards achieving net zero emissions in the near future from all the sectors. The authors have done a commendable job in collecting panel data from various sources, carrying out global analysis and presenting the results with attractive graphics. However, there are some key issues that need to be clarified in the paper for the analysis to be justified. Hence, a major revision has been recommended for this paper to be acceptable for publication.

The supplementary materials and data for the main results through the provided link (<https://zenodo.org/record/7873900#.ZEs28M5Bwk0>) are not accessible. It denies access with a message that the data is under an embargo period until 2024. Hence, this has limited the review process and also the understanding of the overall research design, implementation, results and discussion. It is suggested that the authors investigate this issue and have it sorted out for credibility of the research.

Methods

There are some fundamental concerns in the methods. Although the applied calculations are simple empirically based, a lot of questions arise in the adopted values of the coefficients and estimations, for example, the feasibility and contributions estimations of the different irrigation systems (Eq 4-8). These estimations have a large variation across the geographical regions, continents and countries. Hence, it is suggested that a careful estimation and logical validation on the use of such figures is added to the paper.

IEA reports that "... in 2020, hydropower supplied 17% of global electricity generation, the third largest source after coal and natural gas. Hydropower's contribution is 55% higher than nuclear's and larger than that of all other renewables combined including solar PV, wind, bioenergy and geothermal..." Moreover, it has a very low carbon footprint ~23 gCO₂/kWh compared to solar (~44 gCO₂/kWh). Moreover, a major share of the electricity in Europe and the Americas (which constitute a large part of the current study, Figure 5) is contributed by hydropower. The share was even larger during 2000-2010 (period of analysis of this study). But, very interestingly, this paper has completely ignored hydropower as a low-carbon energy technology and not included it in any of its analyses including the future mitigation studies. This seems to be a significant flaw in the methodology that needs to be addressed critically. With the contribution of hydropower among renewables still remaining intact in the foreseeable future, it is necessary to include hydropower for carbon emission reduction in one of the prominent low-carbon scenarios. If there is a concrete reason for excluding hydropower, it should be explicitly mentioned. The 'current electricity' scenario is not well defined – it is not mentioned what constitutes the energy mix of this scenario. Moreover, a generation mix consisting of renewables could be a more realistic future scenario compared to having scenarios isolating the renewable technologies.

It has been mentioned (L276-278) that the irrigation area has been extracted from two global datasets. However, the authors further mention that the FAOSTAT data also includes irrigated land. Hence, there is a confusion to the readers whether the irrigation land is double counted when data from the two sources are combined. The irrigated land identified from the two sources do not most likely overlap in all the countries. Therefore, if the double counting has been avoided, it needs to be explicitly mentioned how it was done. This is important because the entire analysis and reporting of the research has been done on per unit area energy intensity. Thus, alterations in the irrigated area considerably impacts the results. It is reported (L429-430) that the irrigation water requirements have been estimated using 26 crops using WATNEEDS model. However, it is not clear how the authors arrived at this number. It is understood that the crops and the area they are cultivated in vary differ by countries. The scatterplots presented in the annex (pg 47 in the pdf file) shows a lot of variation,

but these have not been discussed in the results or the discussion section.

It is stated in L258-262 that "... carbon emissions from irrigation accounts for only 1% of global food system emissions and the energy consumption from irrigation accounts for only 2% of global electricity use..." Aren't these statements undermining the findings of this research? A question would immediately come to the readers – why conduct research and propose interventions in an area which would only be able to mitigate 1% of emissions and 2% electricity use? This could lead to the notion that it is not worthwhile trying to reduce energy and carbon emissions from irrigation, rather other areas which continue to contribute much larger share 98-99% need to be targeted. Moreover, the authors move on to write in L260-261 that the energy and emissions from irrigation are not trivial, which contradicts their previous statement. In L262-264, the authors mention that irrigation energy consumption exceeds total agricultural energy consumption in some countries – how is this possible? Isn't agricultural energy consumption inclusive of irrigation energy consumption? Hence, it is strongly suggested that the message the authors are trying to communicate is mentioned clearly and succinctly.

The authors mention that they have used Monte Carlo simulation to arrive at the mean values of energy consumption and carbon emissions (for example in Figure 4) and uncertainty analysis. But the details of the simulation have not been provided. The adopted objective functions and boundary conditions have not been described. This raises a point of doubt among the readers in the simulation generated values.

The authors have provided references to past studies such as Fan et al, Siebert et al, Jagermeyr et al, etc. from which the different datasets have been derived. However, they fail to provide the validation results of these studies. Since the analysis is completely reliant on these datasets, therefore, it would be meaningful if the authors could also prove that these datasets have been validated using ground data to increase the credibility of their work. It is understandable that the different datasets are available at different spatial resolutions. Is there any particular reason why the datasets were resampled to 5 arc-minutes (~9.5 km) (L345-346)? How could the results be impacted if this resolution was altered? It would be helpful particularly to the decision makers of a particular country or regions if the impact of resolution of the data on the results could be explained. Global results could be useful for the academia/research community but it is too coarse and is of little importance to the country level implementations.

Results corresponding to the 'Effects of sustainable irrigation expansion' (L478-483) are not clearly presented in the paper. Similarly, the authors claim that their estimations are consistent with other studies (L497-98), it would be meaningful to also provide the estimates from these other studies. Additionally, it has been mentioned that 'overall efficiency of diesel and electric pumps' are the major sources of uncertainty. Again, the findings from the Monte Carlo simulation have not been discussed in the context of the paper.

The authors claim that they have arrived at the carbon emission intensity from electricity production from an extensive literature (L410-415). It is strongly felt that this needs to be discussed more.

Additionally, the mean values/coefficients/emission factors for use of petroleum, natural gas etc. used in the paper might not be indicative of all the countries. Careful discussion on these aspects is also missing here.

Results

Figure 6 and its description is to be presented in the results section. Moreover, the bar diagram is relatively straight forward, however the percentage values shown in the pie-chart are confusing. They do not add up to 100% and it difficult to interpret. It is suggested that the authors consider this critically. In L231, is the reduction due to a combination of drip and low-carbon electricity? If so, it is suggested to mention the exact combination of the scenarios because it is confusing to the readers as the paper deals with a number of different scenarios.

Discussion

The discussion section seems to be very lightly written given the effort that has been applied to the other sections (particularly the analysis and presentation) of the paper. There are many places where the message is not delivered clearly. For example, the authors argue that water saving technologies

such as sprinkler and drip are more energy demanding. However, it is not clear from the discussion how a rational trade-off can be achieved in this regard. Additionally, the argument that higher irrigation efficiency rarely reduces water consumption (L304-305) seems to be an understatement. Isn't it logical to want to increase the irrigation (command) area or cropping intensity if water is saved and can be used for other purposes?

It is not clear how the authors arrived at different estimations such as the irrigation energy intensity values of 4 GJ/ha and energy required to produce and transport fertilizers (3 GJ/ha) and carbon emission intensity (415 kg CO₂/ha) (L266-275). Moreover, the values presented in L280-288 do not correspond with Figure 6. The authors mention "stringent low-carbon electricity targets" in L289-90. However, they fail to explain what such stringent targets are. Similarly, it is not clear how the authors got the figures presented in L292-296 that have been referred to Supplementary materials Fig 20a (which is not accessible). Moreover, these paragraphs (L266-296) are results and is strongly suggested to be moved to the results section.

There are instances in which the authors report using estimates dating back to the 2000s. It is understandable that these values are considered appropriate for use in the analysis from 2000-2011. However, the paper does not discuss at any point in the paper how these values have changed in the last decade and what impacts they could have on the energy use and emissions. Although a quantitative comparison could be difficult, it would be extremely meaningful if the authors, based on their expertise, shed some light on this as we are almost one and a half decades from the analysis period of this paper where the energy and emission conditions have changed considerably. Moreover, the discussion needs to be directed towards some concrete points that can be derived from the results of this research.

General Presentation

Although the paper is relatively well-structured, the language needs considerable refinement. There are incomplete/grammatically incorrect sentences for example L486-489. There are some sentences which are not clear, for example L234-235. There are a number of repetitions among different sections of the paper, for example, L234-245, L321-325 and L333-336.

The figures and maps presented at the end of the pdf file are not numbered and their caption is missing; some figures are repeated a number of times. In many instances, the units are missing and their references in the main text is not clear. It is suggested that the authors take utmost care in sorting these issues.

Some highly relevant papers are missing. Some of them are given below.

- Energy and water tradeoffs in enhancing food security: A selective international assessment, Energy policy, 2009; 37 (9), 3635-3644
- Climate change and water security: estimating the greenhouse gas costs of achieving water security through investments in modern irrigation technology, 2013; Agricultural Systems 117, 78-89
- Carbon smart agriculture: An integrated regional approach offers significant potential to increase profit and resource use efficiency, and reduce emissions; 2021, Journal of Cleaner Production 282, 124555
- Climate change, water security and the need for integrated policy development: the case of on-farm infrastructure investment in the Australian irrigation sector, 2013, Environmental Research Letters 7 (3), 034006

Reviewer #3:

Remarks to the Author:

I enjoyed reading this article very much. The authors have focused on an important issue, energy and carbon use for irrigation globally, and crafted some beautiful graphics to illustrate their analytical findings. Overall, I thought the manuscript was well written, the scenarios were compelling and interesting, and I applaud the authors for the steps they took to create global estimates with this

physics-based method.

I am going to identify myself in this review, because I will be citing some of my group's work. My name is Anthony Kendall, I am a hydrogeology professor at Michigan State University. In 2020, my master's student Ben McCarthy, published what I believe to be the first large-scale physics-based analysis of energy and carbon emissions from irrigation (McCarthy et al. 2020), focused on the US High Plains Aquifer in Kansas. My expertise in this subject is derived from the years I spent working on this analysis with Ben, in direct coordination with my co-authors including Annick Anctil, a professor of environmental engineering specializing in life cycle analyses.

Before publication, I believe that the authors should enhance their methodology as described below. Without this, their methods are capable of making global maps but can be substantially incorrect (as is in the case in the US). I understand the need to simplify methodology at the global scale, but their conclusions really depend entirely on their methods being accurate. Furthermore, it's not clear that the errors introduced by their methodological oversights are random. In particular, groundwater energy use would appear to be understated--in some cases significantly.

I have three primary issues, and several smaller ones, that I believe should be addressed before further consideration of this manuscript occurs.

Primary Issues:

Calculation of Direct Energy Requirements

First, the authors' calculation of energy requirements for groundwater pumping omits several important factors that can play an important role in accurately computing energy use. The components of the energy required to lift water from the inside of the well bore to the irrigation nozzles/drip outlets is actually calculated as:

$E_{tot} = \rho * g * V * h_{tot} * \eta_p$; where ρ = the density of water, g = the gravitational constant, V = the volume pumped, η_p = the combined pump and prime mover efficiency, and h_{tot} = the total lift height, which is defined as:

$h_{tot} = h_{lift} + h_{press} + h_{loss}$; where h_{lift} = the lift height, h_{press} = the pressurization in the pipe for each system, and h_{loss} = primarily pump losses.

So far, the authors and I agree on these definitions, though our equations are somewhat different (due to units in the total direct energy, mostly). Where my issue arises is in the h_{lift} component. Because the height that water needs to be lifted isn't just the depth to the average regional water table, but rather the sum of the depth to the regional water table, the additional depth of the drawdown cone formed around each well as the pumping season progresses, and the additional depth of water inside the well bore caused by friction in the well screen and well packing material. Thus,

$h_{lift} = h_{wt} + h_{dd} + h_{we}$; where h_{wt} = the water table lift, h_{dd} = the depth of the drawdown cone (time-varying), and h_{we} = depth of water in the well below that of the area immediately outside it.

The authors' omission of the h_{dd} and h_{we} terms can be highly significant, particularly for large irrigated farms with high pumping flow rates. In particular, for areas with shallow water tables, these terms can dominate the h_{lift} . Calculating these two terms is not incredibly difficult, at least if we make simplifying assumptions. See McCarthy et al. (2020) and its supplementary information for details.

"Pump" vs. "Prime Mover" efficiency

One of the reasons that the authors encountered such a wide range of so-called "pump efficiency" estimates is due to the fact that what we might colloquially call "pump efficiency" is actually two things: true pump efficiency, including impeller, driveshaft, and other losses, and; prime mover efficiency, or the efficiency that the electric, diesel, natural gas, or other motor converts its energy source to mechanical movement. Pump efficiencies are usually quite high, on the order of 70% or so for a well-maintained system. Prime mover efficiencies can be much lower, on the order of 25-30% for diesel and natural gas systems, and 80% or so for small electric motors (running on single-phase AC), to 90% or more for large electric motors (using three-phase AC). Thus, total efficiencies are approximately 20% for diesel-driven pumps, and 56% for small electric/63% for large electric pumps. Clarity on this point would help bolster the credibility of the authors' calculations.

Lack of natural gas for the US

Based on the Supplementary Table 3, the US has at least 36% of its pumping provided by sources other than diesel- or electrically-driven pumps. Their citation for this is also woefully out-of-date, being nearly 40 years old. The US government publishes statistics on this, at the county level, every five years through its US Department of Agriculture Irrigation and Water Management Surveys. Now I realize that the US is just one country, but it is the second-largest consumer of energy and emitter of carbon for irrigation, according to these authors. It is also the place where the most study has gone into this issue (more on that below), and the best statistical data are available to validate their approach.

I am also wondering if the authors took the proportion of diesel and electric pumps reported in their table S3 and divided by the sum, to scale the pump proportions to 100%. I assume so, but this is not explicitly stated that I could tell.

Lack of direct emissions of CO₂ from groundwater due to degassing

When groundwater is pumped, it may be supersaturated in carbonate relative to atmospheric pressure. The excess carbonate will then degas, producing direct CO₂ emissions at the irrigated field. This is explained in Wood and Hyndman (2017). McCarthy et al. (2020) show that direct CO₂ emissions from groundwater irrigation can be 30% or more of total CO₂ emissions in regions where groundwater is being actively depleted (as is true in most of the groundwater-irrigated areas worldwide). Inclusion of this process would not be excessively difficult, and simplifying assumptions could be used to produce global estimates.

Disagreement of energy use estimates in the US

In their Supplement, section 2.1, they compare their estimates to those from Liu et al. 2016, and find a 0.83 R² fit between available countries. However, this is entirely driven by their general agreement for India--pulling this statistic much higher. In particular, the second highest energy user, the US, is dramatically overestimated in their study. This would be acceptable if not for the fact that there are other estimates of energy use for the US. In 2022, Sowby and Dicaldo (2022) published an energy estimate for the US that was 219 PJ/yr. Here, the authors estimate 418 PJ/yr. The source the authors' compare to, Liu et al., in their Supplementary Table 2, estimate this total at 141 PJ/yr. Sowby and Dicaldo point to other independent estimates for the US Department of Energy that put the total energy use squarely within the range of 200 PJ/yr, not 400+

Again, the US is just one country, but it is: 1) the second-largest consumer of energy for irrigation, and 2) the one with the best publication record and energy statistics. While it is an outlier in several respects, including the degree of industrialization of its irrigated agricultural sector, and the size of farms irrigated, getting their methods right in that country seems very important. Note that Pakistan also seems to be substantially overestimated.

Also, presenting this validation figure in a log scale would be much more informative across the very wide range of energy consumption.

Smaller Issues:

Feasibility of drip irrigation for soybean

At least in the western hemisphere, soybean is grown on massive fields that are not feasible (at least in the near future) to use drip irrigation. The citation the authors use in their Supplementary Table 5 points to a source that has a table stating that Soybean is feasible to use with drip. However, that table also lists maize as unsuitable for sprinkler irrigation, when we know very well that millions of ha of maize are irrigated with pivot and linear move sprinkler systems worldwide.

Missing References

Here are the three papers I have referenced above. I have no doubt there are others as well. A more thorough literature search would certainly turn up additional contributions.

- McCarthy et al. 2020, Trends in Water Use, Energy Consumption, and Carbon Emissions from Irrigation: Role of Shifting Technologies and Energy Sources
- Sowby and Dicataldo, 2022, The energy footprint of U.S. irrigation: A first estimate from open data
- Wood and Hyndman, 2017, Groundwater Depletion: A Significant Unreported Source of Atmospheric Carbon Dioxide

Over-complexity of primary manuscript graphics

As I said above, the manuscript graphics are well crafted, and striking. However, each one is a unique graphical theme, requiring quite a bit of time to parse. Please try to simplify. I can tell that the influence of the IPCC graphics is strong here, but papers are not IPCC reports. The simpler the graphics, the clearer the story, and the higher the impact a paper can have.

A few examples:

- In Figure 3, having the country in the Sankey diagram twice is quite difficult to use. Their supplemental figure 12 is a much nicer Sankey diagram, actually.
- In Figure 4, there are so many adornments that seem unnecessary. The arrow seems to point in the direction of the bar, which is not needed. The triangles are present in some bars but not all, and sometimes seem identical to the error bars, other times different.
- Figure 5 has a whole collection of themes (radar, pie, bar) and colors.
- I still haven't figured out the mysterious color wheel in each panel of Figure 6.

Other small graphical issues

- In Figure 1, the dashed line is a 1:1 line, but there is no basis for there to be a 1:1 fit between water intensity and energy/CO2 intensity. Instead, a linear fit would then help the reader distinguish between countries above or below median energy/CO2 intensity
- In Figure 2b, the orange bars fade to dark at low values, while there is either no fade, or it is reversed and barely visible in 2a

Response to Reviewers

Reviewers' comments:

Reviewer # 2:

1. This is a very interesting paper that analyses the irrigation related energy consumption and carbon emissions globally. Additionally, the paper quantifies the energy and carbon savings that can be achieved through adoption of low-carbon irrigation interventions. This paper carries significant importance in the context when the whole world is directed towards achieving net zero emissions in the near future from all the sectors. The authors have done a commendable job in collecting panel data from various sources, carrying out global analysis and presenting the results with attractive graphics. However, there are some key issues that need to be clarified in the paper for the analysis to be justified. Hence, a major revision has been recommended for this paper to be acceptable for publication.

Response: Thank you very much for taking your valuable time to review this manuscript. In addition, we really appreciate all your valuable comments and suggestions. Your constructive comments have greatly helped to enhance the clarity and logic of manuscript, make up for the shortcomings in carbon emissions mitigation scenarios, and strengthen the discussion of this study. According to your suggestions, we have revised and improved the manuscript. Next, we make a point-by-point response.

2. The supplementary materials and data for the main results through the provided link (<https://zenodo.org/record/7873900#.ZEs28M5Bwk0>) are not accessible. It denies access with a message that the data is under an embargo period until 2024. Hence, this has limited the review process and also the understanding of the overall research design, implementation, results and discussion. It is suggested that the authors investigate this issue and have it sorted out for credibility of the research.

Revised manuscript: Data availability section

Response: We are very sorry that the link is unavailable due to our negligence. In the revised manuscript, we have attached an updated link to have a better understanding of the design and implementation of this study, and more importantly to aid the review process. The main resulting output datasets in our study are publicly available in the link: <https://doi.org/10.5281/zenodo.10118986>

3. There are some fundamental concerns in the methods. Although the applied calculations are simple empirically based, a lot of questions arise in the adopted values of the coefficients and estimations, for example, the feasibility and contributions

estimations of the different irrigation systems (Eq 4-8). These estimations have a large variation across the geographical regions, continents and countries. Hence, it is suggested that a careful estimation and logical validation on the use of such figures is added to the paper.

Revised manuscript: Supplementary Discussion section 2.3

Response: We agree with the reviewer's opinion. According to the Reviewer's suggestion, in our revised manuscript, we added a logical discussion of feasibility and contributions estimations for drip irrigation and low-carbon electricity in the Supplementary discussion section 2.3. Details are as follows:

(1) ***“Feasibility and contributions estimations for drip irrigation system***

The feasibility of irrigation system (drip, sprinkler or surface irrigation) depends on crop types^[1]. In our study, we first judged the suitability of 26 crops classes for drip irrigation system (Supplementary Table 6). We used the ratio of crops irrigation water consumption applicable to drip irrigation to the total irrigation water consumption of 26 crops to reflect the maximum application of drip irrigation in a country. We used the WATNEEDS model^[2] to estimate irrigation water consumption for 26 crops at a resolution of 5 arcminutes and provided validation of the estimates (Supplementary method section 1.7). However, the feasibility of drip irrigation systems in terms of energy and CO₂ emissions reductions should be the maximum application of drip irrigation minus the proportion of current drip irrigation. Therefore, potential contribution of drip irrigation can be calculated as the product of the feasibility of drip irrigation and the contribution to CO₂ emissions reduction from an increase in the proportion per unit of drip irrigation under the drip irrigation scenario. Although the feasibility analysis of drip irrigation has a more realistic significance in terms of energy and CO₂ emissions reduction, we do not consider the impact of crop structure adjustment in the future. Moreover, in some countries, switching from gravity to drip irrigation does not reduce energy consumption and CO₂ emissions, but has only water-saving benefits.”

(2) ***“Feasibility and contributions estimations for low-carbon electricity***

The feasibility of low-carbon electricity, that is, how much the share of low-carbon electricity can increase by 2050 compared with 2000-2010, determines how much CO₂ emissions can be reduced by 2050. Likewise, potential contribution of low-carbon electricity can be calculated as the product of the feasibility of low-carbon electricity and the contribution to CO₂ emissions reduction from an increase in the proportion per unit of low-carbon electricity under low-carbon electricity scenario. Since information on the proportion of low-carbon electricity by 2050 was missing for each country, we used regional low-carbon electricity targets^[3] as an alternative. However, this resulted in the share of low-carbon electricity in 2000-2010 exceeding the projected regional

targets by 2050 for 37 countries (Afghanistan, Albania, Angola, Armenia, Bhutan, Brazil, Burundi, Cameroon, Canada, Colombia, Congo, Costa Rica, Democratic Republic of Congo, Ethiopia, France, Georgia, Ghana, Guyana, Kyrgyzstan, Laos, Lesotho, Lithuania, Malawi, Mozambique, Namibia, Nepal, Norway, Paraguay, Russia, Sweden, Switzerland, Tajikistan, Tanzania, Uganda, Ukraine, Uruguay, and Zambia), which may ignore the potential contributions of some of these countries to CO₂ emissions reduction. It is remarkable that 68% of these countries have a low-carbon electricity share of over 85% or even close to 100% in 2000-2010^[4], meaning these countries have little contribution to reduce CO₂ emissions of irrigation by 2050.”

Despite uncertainties in feasibility and contribution estimates, we have done our best to provide a more realistic reference for future energy consumption and CO₂ emissions reductions of irrigation.

References:

- [1] Jägermeyr J, Gerten D, Heinke J, et al. Water savings potentials of irrigation systems: global simulation of processes and linkages[J]. *Hydrology and Earth System Sciences*, 2015, 19(7): 3073-3091.
- [2] Chiarelli D D, Passera C, Rosa L, et al. The green and blue crop water requirement WATNEEDS model and its global gridded outputs[J]. *Scientific data*, 2020, 7(1): 273.
- [3] Enerdata. Global energy & climate outlook 2050.
<https://eneroutlook.enerdata.net/forecast-world-co2-intensity-of-electricity-generation.html>
- [4] Ritchie, H. & Roser, M. Electricity mix. Our World in Data.
<https://ourworldindata.org/electricity-mix>

4. IEA reports that “... in 2020, hydropower supplied 17% of global electricity generation, the third largest source after coal and natural gas. Hydropower’s contribution is 55% higher than nuclear’s and larger than that of all other renewables combined including solar PV, wind, bioenergy and geothermal...” Moreover, it has a very low carbon footprint ~23 gCO₂/kWh compared to solar (~44 gCO₂/kWh). Moreover, a major share of the electricity in Europe and the Americas (which constitute a large part of the current study, Figure 5) is contributed by hydropower. The share was even larger during 2000-2010 (period of analysis of this study). But, very interestingly, this paper has completely ignored hydropower as a low-carbon energy technology and not included it in any of its analyses including the future mitigation studies. This seems to be a significant flaw in the methodology that needs to be addressed critically. With the contribution of hydropower among renewables still remaining intact in the foreseeable future, it is necessary to include hydropower for carbon emission reduction in one of the prominent low-carbon scenarios. If there is a concrete reason for excluding hydropower, it should be explicitly mentioned. The ‘current electricity’ scenario is not

well defined – it is not mentioned what constitutes the energy mix of this scenario. Moreover, a generation mix consisting of renewables could be a more realistic future scenario compared to having scenarios isolating the renewable technologies.

Revised manuscript: Methods in main text Line 519-526; Supplementary Methods 1.6

Response: We agree with the reviewer’s opinion. According to the Reviewer’s suggestion, in our revised manuscript, we included hydropower as a low-carbon energy technology in our analysis of mitigation scenarios. Based on electricity mix by 2050 reported by IEA, we calculated the carbon mitigation potential under a mix electricity scenario. In addition, the current electricity is well defined in the caption of Fig. 4. Details are as follows:

(1) Carbon footprint of hydropower

According to Ubierna et al. (2022), global median lifecycle greenhouse emissions of hydropower is 23 g CO_{2e}/kWh (5~99 g CO_{2e}/kWh). The value is aligned with the IPCC estimates ^[1]. “*The carbon footprint of solar, wind, nuclear, and hydropower based on lifecycle assessment is 44, 11, 12, and 23 g CO₂/kWh, respectively ^{[1],[2]}.*”

(2) “**Share of low-carbon electricity by 2050**

According to the IEA net-zero by 2050 roadmap, to achieve the net-zero GHG emissions by 2050, solar, wind, hydropower, and nuclear would provide 23469 TWh, 24785 TWh, 8461 TWh, and 5496 TWh of electricity, accounting for 33%, 35%, 12% and 8% of total electricity generation, respectively ^[3]. In this study, we assumed that energy for irrigation under a mixed electricity scenario by 2050 is composed of the above four power sources and is distributed in proportion to the electricity generation. Subsequently, we calculated the carbon footprint of mixed electricity based on the carbon footprint as well as electricity generation for solar, wind, hydropower, and nuclear.”

(3) **Current electricity**

The current electricity represents the electricity production structure during 2000-2010, where the electricity comes from fossil fuels, nuclear, hydro, geothermal, solar, wind, tide, wave, ocean and biofuels ^[3]. The IEA report provides carbon footprint of current electricity mix for 2000–2010^[3]. According to the Reviewer’s suggestion, in our revised manuscript, we used 2000-2010 electricity mix instead of current electricity.

The figure 4 is shown as below:

Fig. 4 Mitigation potentials of energy and CO₂ emissions under different scenarios. **a** The sprinkler scenario and drip scenario have different water-use intensities. For the electric pumping scenario, the energy consumption of irrigation does not change with the CO₂ intensity of electricity. **b** In sprinkler and drip scenarios, CO₂ intensity of energy (electricity during 2000–2010) remains unchanged. In electric pumping scenarios, electricity comes from current electricity mix, solar, wind, nuclear, hydropower, or a mix of the four low-carbon electricity according to IEA net-zero by 2050 projections^[3]. The baseline scenario reflects energy consumption and energy-related CO₂ emissions during 2000–2010. The 2000-2010 electricity mix represents the electricity production structure during 2000–2010, where the electricity comes from fossil fuels, nuclear, hydro, geothermal, solar, wind, tide, wave, ocean and biofuels^[4].

References:

- [1] Ubierna M, Santos C D, Mercier-Blais S. Water security and climate change: hydropower reservoir greenhouse gas emissions[J]. *Water Security Under Climate Change*, 2022: 69-94.
- [2] UNECE. Carbon neutrality in the UNECE region: integrated life-cycle assessment of electricity source. (Geneva: United Nations, 2022). (Page 82)
https://unece.org/sites/default/files/2022-08/LCA_0708_correction.pdf
- [3] IEA. Net zero by 2050: A roadmap for the global energy sector. 2021. (Page 199)
https://iea.blob.core.windows.net/assets/deebef5d-0c34-4539-9d0c-10b13d840027/NetZeroBy2050-ARoadmapfortheGlobalEnergySector_CORR.pdf
- [4] IEA. CO₂ Emissions from Fuel Combustion - 2012 Highlights. 2012. (Page 113)
<https://www.osti.gov/etdweb/servlets/purl/22083097>

5. It has been mentioned (L276-278) that the irrigation area has been extracted from two global datasets. However, the authors further mention that the FAOSTAT data also

includes irrigated land. Hence, there is a confusion to the readers whether the irrigation land is double counted when data from the two sources are combined. The irrigated land identified from the two sources do not most likely overlap in all the countries. Therefore, if the double counting has been avoided, it needs to be explicitly mentioned how it was done. This is important because the entire analysis and reporting of the research has been done on per unit area energy intensity. Thus, alterations in the irrigated area considerably impacts the results.

Revised manuscript: Methods in main text Line 575-577; Supplementary Table 8

Response: We apologies for the confusion to readers. As the reviewer said, the total energy input and CO₂ emissions from fertilizers, machinery and fuels are calculated based on energy and CO₂ emissions intensity per unit area. In the previous manuscript, the reason for using two global datasets is that the energy input and corresponding CO₂ emissions from fertilizers, machinery and fuels on irrigated land are the product of its energy intensity and irrigated area. However, fertilizer, machinery, and fuel are not just used to irrigated land. Therefore, the total energy inputs and CO₂ emissions from fertilizers, machinery and fuels in agriculture should be calculated as its energy intensity multiplied by total cropland area.

According to the Reviewer's suggestion, to avoid confusion for readers, we only calculated the total energy inputs and corresponding CO₂ emissions from fertilizers, machinery and fuel in total cropland in our revised manuscript. In this study, we deliberately highlighted the high energy consumption of irrigation by contrasting with other energy inputs, and weaken the discussion of other energy input on the farm. Estimates of energy inputs and corresponding carbon emissions from other farm operations did not impacts the results of this study. In addition, we have checked the whole article to make sure that such problems do not arise.

6. It is reported (L429-430) that the irrigation water requirements have been estimated using 26 crops using WATNEEDS model. However, it is not clear how the authors arrived at this number. It is understood that the crops and the area they are cultivated in vary differ by countries. The scatterplots presented in the annex (pg 47 in the pdf file) shows a lot of variation, but these have not been discussed in the results or the discussion section.

Revised manuscript: Supplementary Methods section 1.7

Response: As the reviewers said, the crops and the area they are cultivated in vary differ by countries. In this study, we used the MIRCA 2000 dataset ^[1] (5 arc-minutes resolution), which provides a global growing area grid (unit: hectare) of 26 irrigated crop classes varying differ by countries, to estimate irrigation water requirements of crops. The figure below shows the spatial distribution of cultivated area for two irrigated crops (cotton and maize) (Figure. 1).

In this study, the irrigation water requirements for each crop were mainly used to analyze the feasibility of drip irrigation in each country. Therefore, the estimation of irrigation water requirements is not introduced in detail in the main text. In the Supplementary Methods section 1.7, we give a detailed instructions for the estimation of irrigation water requirement.

Figure. 1 Spatial distribution of cultivated area for two irrigated crops (top: cotton; bottom: maize).

References:

[1] Portmann F T, Siebert S, Döll P. MIRCA2000—Global monthly irrigated and rainfed crop areas around the year 2000: A new high-resolution data set for agricultural and hydrological modeling[J]. *Global biogeochemical cycles*, 2010, 24(1).

The datasets can be download from the following link:

https://www.uni-frankfurt.de/45217893/4_Monthly_irrigated_and_rainfed_crop_areas

7. It is stated in L258-262 that “... carbon emissions from irrigation accounts for only 1% of global food system emissions and the energy consumption from irrigation accounts for only 2% of global electricity use...” Aren’t these statements undermining the findings of this research? A question would immediately come to the readers – why conduct research and propose interventions in an area which would only be able to mitigate 1% of emissions and 2% electricity use? This could lead to the notion that it is not worthwhile trying to reduce energy and carbon emissions from irrigation, rather

other areas which continue to contribute much larger share 98-99% need to be targeted. Moreover, the authors move on to write in L260-261 that the energy and emissions from irrigation are not trivial, which contradicts their previous statement.

Revised manuscript: Discussion section in main text

Response: According to the Reviewer's suggestion, in our revised manuscript, we have removed these inappropriate and contradictory statements. In addition, we rewrote the discussion section in response to our findings.

Our revised discussion focuses on the following aspects: (1) A series of adverse chain reactions caused by irrigation with high energy input and CO₂ emissions; (2) analysis of trade-off between water savings and reductions of energy and CO₂ emissions, and suggestions; (3) the potential and challenge of low-carbon electricity in CO₂ emissions reduction; (4) changes in energy consumption and CO₂ emissions irrigation in recent decade years, and implications of this work.

8. In L262-264, the authors mention that irrigation energy consumption exceeds total agricultural energy consumption in some countries – how is this possible? Isn't agricultural energy consumption inclusive of irrigation energy consumption? Hence, it is strongly suggested that the message the authors are trying to communicate is mentioned clearly and succinctly.

Revised manuscript: ---- Supplementary Fig. 20

Response: According to the Reviewer's suggestion, we first checked the definition of total final energy consumption of agricultural in IEA ^[1]. According to database documentation of IEA, Agriculture/forestry includes deliveries to users classified as agriculture, hunting and forestry by the International Standard Industrial Classification (ISIC), and therefore includes energy consumed by such users whether for traction (excluding agricultural highway use), power or heating (agricultural and domestic) [ISIC Rev. 4 Divisions 01 and 02]. In addition, we also check ISIC Rev. 4 Divisions 01 and 02 ^[2] and found that irrigation is not included. A more specific examples, according to IEA database, total final energy consumption of agriculture in Pakistan is 37 PJ (maximum value during 2000-2010) in 2009. However, current estimates (Liu et al (2016): 75 PJ; Siyal and Gerbens-Leenes (2022): 103PJ) of energy consumption for irrigation in Pakistan all exceed the value, which indirectly indicates that the IEA does not include irrigation energy consumption in agricultural final energy consumption.

In addition, we are the first to quantify global energy use from irrigation. Our study can fill the gaps in current agricultural systems regarding energy use.

To avoid causing confusion to the reader, in our revised manuscript, we use the ratio of irrigation energy consumption to the total energy supply to reflect the pressure of irrigation energy consumption on energy supply (Supplementary Fig. 20).

Supplementary Fig. 20. Ratio of energy use from irrigation in total energy supply. **a** Current energy use of irrigation. **b** Energy use of irrigation under current plus sustainable irrigation expansion of 3 °C climate in 2050. Total country-level energy supply data is derived from IEA during 2000–2010. The grey areas represent missing data in IEA dataset. Here, an important reason for the comparison between energy consumption of irrigation and national energy supply is that according to the definition and classification of agricultural final energy consumption in IEA, energy consumption of irrigation is not included.

References:

- [1] IEA. Database documentation. (Page 16)
https://iea.blob.core.windows.net/assets/25266100-859c-4b9c-bd46-cc4069bd4412/WORLDBAL_Documentation.pdf
- [2] International Standard Industrial Classification of All Economic Activities (ISIC), Rev.4. (Page 60)
https://unstats.un.org/unsd/publication/seriesm/seriesm_4rev4e.pdf
- [3] Liu Y, Hejazi M, Kyle P, et al. Global and regional evaluation of energy for water[J]. Environmental Science & Technology, 2016, 50(17): 9736-9745.

[4] Siyal, A. W. & Gerbens-Leenes, P. W. The water–energy nexus in irrigated agriculture in South Asia: Critical hotspots of irrigation water use, related energy application, and greenhouse gas emissions for wheat, rice, sugarcane, and cotton in Pakistan. *Front. Water*. 4, 941722 (2022).

9. The authors mention that they have used Monte Carlo simulation to arrive at the mean values of energy consumption and carbon emissions (for example in Figure 4) and uncertainty analysis. But the details of the simulation have not been provided. The adopted objective functions and boundary conditions have not been described. This raises a point of doubt among the readers in the simulation generated values.

Revised manuscript: Supplementary Methods section 1.2 and Discussion section 2.1

Response: We are very sorry for a mistake on the concept of pumping efficiency which led to a wide range value of pumping efficiency. As suggested by another reviewer with expertise in this subject, pump efficiency reflect the loss of impeller and driveshaft; Prime mover efficiency is the efficiency with which electricity, diesel, natural gas or other motors converts its energy source to mechanical movement. Pump efficiencies are usually quite high, on the order of 70% or so for a well-maintained system. Prime mover efficiencies can be much lower, on the order of 30% for diesel and 25% for natural gas systems, and 80% or so for small electric motors (running on single-phase AC), to 90% or more for large electric motors (using three-phase AC) ^[1]. Thus, total efficiencies are approximately 21% for diesel-driven pumps, 17.5% for natural gas pumps, and 56% for small electric/63% for large electric pumps. In our revised manuscript, we adopted reviewer's # 2 recommendation that total pump efficiency of 56% for electric pumps, 21% for diesel-driven pumps, 17.5% for natural gas pumps for a global analysis.

In our previous manuscript, we collected overall pump efficiency values from a large literature, which has a wide range. Since the collected values conform to the normal distribution, we generate random numbers according to the mean and variance of the overall pumping efficiency values and the conditions that must satisfy the normal distribution. However, the main reason for such a wide range is caused by human factors, such as poorly maintained and failure to select equipment to match the specific pumping conditions ^[2]. Therefore, in our revised version, these human factors were not considered in our study. In our Supplementary Methods section, we supplemented and refined the information on the adoption of pump efficiency.

Based on the revised total efficiency values, we re-estimated energy consumption and CO₂ emissions from irrigation, and compared them with previous studies. Details are as follows:

“In this study, global energy consumption from irrigation was 1896 PJ. According to Liu et al., 2016, the energy consumption from agricultural water source and conveyance was 2433 PJ (Supplementary Table 9), which was based on irrigation

water withdrawal and energy intensity values for each water process and source. From the comparison of energy consumption on the national scale, we noted a significant correlation ($R^2 = 0.68$) between them (Supplementary Fig. 23). However, there were differences in some countries.

In the United States and Pakistan, the energy consumption estimated by this study is 30%-40% higher than the results estimated by Liu et al., 2016 (Supplementary Table 9). The energy consumption of India estimated in this study was 78% lower than that estimated by Liu et al., 2016. However, the results from other studies on energy consumption of irrigation in India, Pakistan, and the United States were consistent with the results of this study (Supplementary Table 9).

Accordingly, in India and China, the energy-related CO₂ emissions estimated in our study were 70 Mt CO₂ and 35 Mt CO₂, which are close to the estimation of 59-92 Mt CO₂ in 2009 by Shah et al., 2009 and 34-47 Mt CO₂ in 2010 by Zou et al., 2015.”

Supplementary Table 9. Comparison of energy consumption and CO₂ emissions from irrigation between this study and previous studies.

Energy consumption (PJ)					CO ₂ emissions (Mt CO ₂)		Source
Global	The United States	China	India	Pakistan	China	India	
2433	141	261	953	75			Liu et al. 2016
	219						Sowby and Dicaldo. 2022
				103			Siyal and Gerbens-Leenes. 2022
			439				Can et al. 2009
						59-92	Shah et al. 2009
					34-47		Zou et al. 2015
1896	205	299	535	135	35	70	This study

Supplementary Fig. 23. Country-level comparison of energy consumption from irrigation between this study and Liu et al., 2016.

References:

- [1] McCarthy B, Anex R, Wang Y, et al. Trends in water use, energy consumption, and carbon emissions from irrigation: role of shifting technologies and energy sources[J]. *Environmental Science & Technology*, 2020, 54(23): 15329-15337.
- [2] New L L. Pumping Plant Efficiency and Irrigation Costs[J]. Leaflet/Texas Agricultural Extension Service; no. 2218., 1986.
<file:///C:/Users/dell/Desktop/%E7%BE%8E%E5%9B%BD%E5%A4%A9%E7%84%B6%E6%B0%94%E6%B3%B5%E7%9B%B8%E5%85%B3%E6%96%87%E7%8C%AE/Pumping%20Plant%20Efficiency%20and%20Irrigation%20Costs.pdf>
- [3] Liu, Y. et al. Global and regional evaluation of energy for water. *Environ. Sci. Technol.* 50, 9736-9745 (2016).
- [4] Sowby R B, Dicataldo E. The energy footprint of US irrigation: A first estimate from open data[J]. *Energy Nexus*, 2022, 6: 100066.
- [5] Siyal A W, Gerbens-Leenes P W. The water–energy nexus in irrigated agriculture in South Asia: Critical hotspots of irrigation water use, related energy application, and greenhouse gas emissions for wheat, rice, sugarcane, and cotton in Pakistan[J]. *Frontiers in Water*, 2022, 4: 941722.
- [6] Can, D L R D., McNeil M, Sathaye J. India energy outlook: end use demand in India to 2020. Lawrence Berkeley National Lab. (LBNL), Berkeley, CA (United States), 2009.
<https://eta-publications.lbl.gov/sites/default/files/lbnl-1751e.pdf>
- [7] Shah, T. Climate change and groundwater: India's opportunities for mitigation and adaptation. *Environ. Res. Lett.* 4, 35005 (2009).
- [8] Zou, X. et al. Greenhouse gas emissions from agricultural irrigation in China. *Mitig. Adapt. Strateg. Glob. Chang.* 20, 295-315 (2015).

10. The authors have provided references to past studies such as Fan et al, Siebert et al, Jagermeyr et al, etc. from which the different datasets have been derived. However,

they fail to provide the validation results of these studies. Since the analysis is completely reliant on these datasets, therefore, it would be meaningful if the authors could also prove that these datasets have been validated using ground data to increase the credibility of their work.

Revised manuscript: ----

Response: We agree with the reviewer's opinion that the reliability of our results depends entirely on the precision of the datasets used. Therefore, we verify the reliability of datasets used in our study from various aspects.

Details are as follows:

(1) Groundwater table depth

First, the global groundwater table depth datasets derived from Fan et al. 2013 are constructed based on 1,603,781 direct well measurements (Fig.1 in Fan et al. 2013) collected from governments archives and published studies. Second, Fan et al. 2013 presented comparisons of observed and modeled groundwater water table depth in terms of area distribution and grid cells (Fig.2 and Fig.S6 in Fan et al. 2013), and results show relatively consistent in spatial distribution. Finally, the datasets, as openly accessible global data, are widely used in hydrological research (Maggi et al.2023; Zhao et al.2023; Huggins et al.2023).

(2) Percentage of surface water and groundwater for irrigation

First, the datasets of source of irrigation water for irrigation (latest version) derived from Siebert et al., 2013 are mainly based on statistics published in national census reports available on-line or made available from the FAO-AQUASTAT library. Second, Siebert et al.2013 present a comparison of the datasets on area equipped for irrigation in version 5 of the Global Map of Irrigation Areas to the most recent statistics in AQUASTAT and FAOSTAT, and results show in general a very good agreement between the data bases (Table B1 in Siebert et al.2013). Finally, this dataset is widely used in irrigated agricultural water analysis (Jha et al. 2022; Al-Yaari et al. 2022; Ruess et al. 2023).

(3) The share of surface, sprinkler, and drip

The country-level datasets of share of surface, sprinkler, and drip derived from Jägermeyr et al. 2015 are mainly compiled from FAO (2014) and ICID (International Commission on Irrigation and Drainage, 2012). Although this data is country-scale resolution, we believe it is the only reliable data available with global coverage.

(4) Irrigation water withdrawal

The datasets of reconstructed global gridded irrigation water withdrawal derived from Huang et al. 2018 is generated based on gridded monthly irrigation water withdrawals

as simulated by four global hydrological models (GHMs). Irrigation water withdrawals simulated by these four GHMs all have reasonable agreement (correlation coefficient, $r > 0.7$) with FAO AQUASTAT and USGS estimates at the country level and US state level, respectively.

Furthermore, irrigation water withdrawals simulated by these four GHMs are corrected by correction factors to adjust the irrigation water withdrawal estimates by GHMs to match the reported data at the country or state level. However, Huang et al. 2018 did not verify the accuracy of the reconstructed datasets, here we provide a country-level comparison of the reconstructed global gridded irrigation water withdrawal datasets used in our study to the statistics in FAO AQUASTAT. The country-level irrigation water withdrawal from FAO AQUASTAT can be found via the link (<https://firebasestorage.googleapis.com/v0/b/fao-aquastat.appspot.com/o/PDF%2FTABLES%2FTable4.pdf?alt=media&token=fdba62dc-ca8f-4b80-adcd-909baa2ddf87>).

Figure.1 A comparison of reconstructed irrigation water withdrawal used in our study to FAO observation.

References:

- [1] Fan Y, Li H, Miguez-Macho G. Global patterns of groundwater table depth[J]. *Science*, 2013, 339(6122): 940-943.
- [2] Maggi F, Tang F H M, Tubiello F N. Agricultural pesticide land budget and river discharge to oceans[J]. *Nature*, 2023, 620(7976): 1013-1017.
- [3] Zhao F, Yang L, Yen H, et al. Reducing risks of antibiotics to crop production requires land system intensification within thresholds[J]. *Nature Communications*, 2023, 14(1): 6094.
- [4] Huggins X, Gleeson T, Serrano D, et al. Groundwatersheds of protected areas reveal globally overlooked risks and opportunities[J]. *Nature Sustainability*, 2023.
- [5] Siebert S, Henrich V, Frenken K, et al. Update of the digital global map of irrigation areas to version 5[J]. Rheinische Friedrich-Wilhelms-Universität, Bonn, Germany and Food and Agriculture Organization of the United Nations, Rome, Italy, 2013.

- [6] Jha R, Mondal A, Devanand A, et al. Limited influence of irrigation on pre-monsoon heat stress in the Indo-Gangetic Plain[J]. *Nature Communications*, 2022, 13(1): 4275.
- [7] Al-Yaari A, Ducharne A, Thiery W, et al. The role of irrigation expansion on historical climate change: insights from CMIP6[J]. *Earth's Future*, 2022, 10(11): e2022EF002859.
- [8] Ruess P J, Konar M, Wanders N, et al. Irrigation by crop in the Continental United States from 2008 to 2020[J]. *Water Resources Research*, 2023, 59(2): e2022WR032804.
- [9] Jägermeyr J, Gerten D, Heinke J, et al. Water savings potentials of irrigation systems: global simulation of processes and linkages[J]. *Hydrology and Earth System Sciences*, 2015, 19(7): 3073-3091.
- [10] Huang Z, Hejazi M, Li X, et al. Reconstruction of global gridded monthly sectoral water withdrawals for 1971–2010 and analysis of their spatiotemporal patterns[J]. *Hydrology and Earth System Sciences*, 2018, 22(4): 2117-2133.

11. It is understandable that the different datasets are available at different spatial resolutions. Is there any particular reason why the datasets were resampled to 5 arc-minutes (~9.5 km) (L345-346)? How could the results be impacted if this resolution was altered? It would be helpful particularly to the of resolution of the data on the results could be explained. Global results could be useful for the academia/research community but it is too coarse and is of little importance to the country level implementations.

Revised manuscript: Methods in main text Line 415-418

Response: That is a very good question. In this study, the impact of data resampling was not analyzed. The original resolution of groundwater table depth datasets and irrigation water withdrawal datasets is 30 arc-second and 30 arc-minutes, respectively. The percentage of surface water and groundwater for irrigation datasets are at a resolution of 5 arc-minutes. To match the data on the proportion of irrigation water source, we chose to resample the irrigation water withdrawal as well as the groundwater table depth datasets. At the grid scale, non-uniform data resolution can hinder calculation between grids, such as 360×720 grids cannot be multiplied with 2160×4320 grids. In addition, we used the nearest neighbor resampling method, which does not change any of the values of cells from the input layer. The value of each cell in an output raster is calculated using the value of the nearest cell in an input raster.

According to the Reviewer's suggestion, to test whether the resampled data affects results, we calculated the irrigation water withdrawal at different resolutions in 2005.

(1) 30 arc-minutes

The irrigation water withdrawal datasets derived from Huang et al. (2018) has a resolution of 30 arc-minutes, and the unit is mm/year. We multiply the irrigation water withdrawal of 30 arc-minutes resolution by cell area of 30 arc-minutes to calculate the irrigation water withdrawal in cubic meters (Figure. 1). The cell area of 30 arc-minutes

resolution can be obtained from Portmann et al. (2010). The cell area is given in hectare (ha), and was computed using an equal-area projection. Subsequently, we divide the values into 10 levels according to the natural break method.

Figure. 1 Irrigation water withdrawal at a resolution of 30 arc-minutes in cubic meters in 2005.

(2) 5 arc-minutes

The irrigation water withdrawal datasets were resampled to a 5 arc-minutes resolution using nearest neighbor method. We multiply the irrigation water withdrawal of 5 arc-minutes resolution by cell area of 5 arc-minutes to calculate the irrigation water withdrawal in cubic meters (Figure. 2). The cell area of 5 arc-minutes resolution can also be obtained from Portmann et al. (2010). Subsequently, we divide the values into 10 levels according to the natural break method.

Figure. 2 Irrigation water withdrawal at a resolution of 5 arc-minutes in cubic meters in 2005.

We found that the spatial distribution of irrigation water withdrawal after resampling based on nearest neighbor method was completely consistent with that without resampling. In addition, the total irrigation water withdrawal has not changed as a result. Therefore, the precision of the analysis depends heavily on the resolution of the original data, and the quadratic effect of resampling based on nearest neighbor method on the

results is very small. In our methods section, we supplement the resampling method used in our study.

We shared our detailed results (raster data with 5 arc-minutes resolution) on Zenodo (see response to comments 2), where they are freely available. Although the 5-arc resolution results are relatively coarse for local areas analyses, our findings provide a reference for the country level implementations.

References:

[1] Huang Z, Hejazi M, Li X, et al. Reconstruction of global gridded monthly sectoral water withdrawals for 1971–2010 and analysis of their spatiotemporal patterns[J]. *Hydrology and Earth System Sciences*, 2018, 22(4): 2117-2133.

[2] Portmann F T, Siebert S, Döll P. MIRCA2000—Global monthly irrigated and rainfed crop areas around the year 2000: A new high-resolution data set for agricultural and hydrological modeling[J]. *Global biogeochemical cycles*, 2010, 24(1).

Cell area can be download from the following link:

<https://zenodo.org/records/7422506>

12. Results corresponding to the ‘Effects of sustainable irrigation expansion’ (L478-483) are not clearly presented in the paper.

Revised manuscript: The third part of the results

Response: According to the Reviewer’s suggestion, in our revised manuscript, we added an analysis of effects of sustainable irrigation expansion in the second part of the results (Global energy consumption and CO₂ emissions), and incorporated energy and CO₂ emissions from sustainable irrigation expansion in Figure 2 using a stacked bar graph. Details are as follows:

Fig. 2 Global energy consumption and CO₂ emissions from irrigation. Country-level energy consumption and CO₂ emissions are based on pixel sum statistics. **a** Energy consumption (PJ per year) under current 2000–2010 and sustainable irrigation expansion of 3 °C warmer climate. **b** CO₂ emissions (Mt CO₂ per year) from energy consumption under 2000-2010 and sustainable irrigation expansion of 3 °C warmer climate. We selected the top 20 countries with the highest energy consumption and CO₂ emissions. The upper right subgraphs represent a summary of energy consumption and CO₂ emissions by regions as well as globally.

“Sustainable irrigation is irrigation practices that do not deplete groundwater stocks and impair freshwater ecosystems. As global warming and food demand increases, sustainable irrigation expansion is an important adaptation solution to future food crises and climate change. The expansion of irrigation inevitably leads to energy consumption and energy-related CO₂ emissions. Therefore, we have also

conducted an estimation of the energy and CO₂ emissions associated with future irrigation expansion. Our assumption is that irrigation will expand in regions where water is expected to be locally available to meet the demand for irrigation water in a climate that is 3 °C warmer—a projected warming level under business-as-usual scenarios⁴. We presuppose that the existing country-specific efficiency of irrigation water usage, encompassing drip, sprinkler, and surface irrigation systems, remains constant in the envisioned scenario of sustainable irrigation expansion in the future. Our estimate indicates that the global additional energy consumption for future irrigation due to sustainable expansion would be 536 PJ, representing 28% of the total irrigation energy consumption in 2000-2010 (Fig. 2a and Supplementary Fig. 7a).

In North America, Africa, and South America, the energy consumption arising from sustainable irrigation expansion is projected to require an additional 139 PJ, 63 PJ, and 60 PJ, respectively, each exceeding 50% of their current energy consumption (Fig. 2a). Notably, Europe anticipates an additional energy consumption from sustainable irrigation expansion of 148 PJ, which is twice its current energy consumption (Fig. 2a). The United States, India, Russia, Brazil, and Mexico are identified as the top countries with the highest energy consumption from sustainable irrigation expansion, contributing 97 PJ, 49 PJ, 39 PJ, 39 PJ, and 18 PJ, respectively, collectively accounting for 45% of the total energy consumption from sustainable irrigation expansion.

Assuming the full adoption of electric pumps and the projected regional carbon intensity of electricity in 2050, the additional energy-related CO₂ emissions resulting from sustainable irrigation expansion are estimated to be 15 Mt CO₂ per year, constituting 7% of the 2000-2010 total energy-related CO₂ emissions (Fig. 2b and Supplementary Fig. 7b). India and Russia emerge as the most significant contributors to CO₂ emissions from sustainable irrigation expansion, emitting 3 and 2 Mt CO₂ per year, respectively.”

13. Similarly, the authors claim that their estimations are consistent with other studies (L497-98), it would be meaningful to also provide the estimates from these other studies.

Revised manuscript: Methods in main text Line 592-594; Supplementary Table 3

Response: We agree with reviewer’s opinion. According to reviewer’s suggestion, we provided share of electric and diesel pumping for eight countries (Australia, Bangladesh, China, India, Iran, Nepal, Pakistan, Uzbekistan) from other studies in our Supplementary Table 3. In addition, supplementary explanations are provided in the main text. In the United States, county-level irrigation pumps data can be obtained from U.S. Department of Agriculture ^[1].

References:

[1] U.S. Department of Agriculture. Farm and Ranch Irrigation Survey (2003). (2004). (Page 68)
<https://nawi.openei.org/files/8/2003%20Farm%20and%20Ranch%20Irrigation%20Survey.pdf>

14. Additionally, it has been mentioned that ‘overall efficiency of diesel and electric pumps’ are the major sources of uncertainty. Again, the findings from the Monte Carlo simulation have not been discussed in the context of the paper.

Revised manuscript: ----

Response: Please see the response to comment (9) for an explanation of pumping efficiency. In addition, as suggested by another reviewer, total efficiencies of pumps are well constrained in our revised manuscript. Therefore, the Monte Carlo simulations are not included in our newly revised manuscript.

15. The authors claim that they have arrived at the carbon emission intensity from electricity production from an extensive literature (L410-415). It is strongly felt that this needs to be discussed more. Additionally, the mean values/coefficients/emission factors for use of petroleum, natural gas etc. used in the paper might not be indicative of all the countries. Careful discussion on these aspects is also missing here.

Revised manuscript: Methods Caveats section

Response: We agree with reviewer’s opinion. In our previous manuscript, we used typical values (Line 410-415 of previous manuscript) of carbon footprints of solar, wind, and nuclear based on life cycle assessment. However, carbon footprints of solar, wind, and nuclear have a wide range. Therefore, we provided a sensitivity analysis (Supplementary Method Section 1.10 and Supplementary Table 7). In addition, we use regional or fixed conversion factors to calculate other energy inputs on farms, which ignores the differences in technology levels between countries. Therefore, according to the Reviewer’s suggestion, in our revised Methods section, we added the uncertainty discussion. Details are as follows:

“When we calculated the mitigation potential of low-carbon electricity to CO₂ emissions, typical values of carbon footprints of solar, wind, nuclear, and hydropower were assumed. However, differences in technology levels between countries and wide ranges of carbon footprints for low-carbon electricity throughout the life cycle led to uncertainties in the mitigation potential of low-carbon electricity on CO₂ emissions of irrigation (Supplementary Table 7). Considering the availability of data, we use regional or fixed conversion factors when estimating energy inputs and corresponding carbon emissions of other farm operations, such as emission factors for the use of liquefied petroleum gas, motor gasoline, and gas-diesel oils, which indirectly affects the proportion of irrigation energy input in the total energy input on farms.”

16. Figure 6 and its description is to be presented in the results section. Moreover, the

bar diagram is relatively straight forward, however the percentage values shown in the pie-chart are confusing. They do not add up to 100% and it difficult to interpret. It is suggested that the authors consider this critically.

Revised manuscript: ----

Response: According to reviewer's suggestion, we added a new result section to analyse the irrigation contribution to on farm. Details are as follows:

“Irrigation contribution to on farm energy use

Based on the analysis of energy intensity and CO₂ emissions intensity of irrigation at the national scale (Fig. 1a,c), we further analyze irrigation contribution to on farm energy use. Figure 6 shows the comparison of energy input and carbon emissions intensity of irrigation with total energy input and carbon emissions intensity of farm in sub-regions worldwide. Globally, energy input intensity of irrigation accounts for 32% of global energy input intensity, and over 50% in the sub-Saharan Africa, Middle East and North Africa, South Asia, and Latin America and Caribbean (Fig. 6a). Accordingly, CO₂ emissions intensity of irrigation accounts for 33% of global CO₂ emissions intensity, and over 50% in sub-Saharan Africa, Middle East and North Africa, and South Asia (Fig. 6b). In addition, the largest energy input intensity and CO₂ emission intensity of North America also come from irrigation.

Based on cropland area from FAOSTAT, we estimate global energy consumption (PJ) and the corresponding carbon emissions (Mt CO₂e) from fertilizers, machinery, and fuel (Supplementary Table 8). We find that energy consumption and carbon emissions from irrigation account for approximately 15% of the total energy consumption and carbon emissions in agriculture (Supplementary Table 8).”

In addition, we are very sorry for the confusing pie-chart. In the previous version, the pie chart represented the comparison of energy intensity on six continents, and the percentage values represented the share of irrigation energy intensity in total energy input intensity for six continents. To avoid causing misunderstanding, in our newly revised version, we removed the analysis of the six continents and retained the more detailed analysis of the nine sub-regions. The improved figure is shown below:

Fig. 6 Contribution of irrigation energy input intensity to total energy input intensity and comparison of CO₂ emissions intensity generated by energy inputs intensity during 2000–2010. **a** Energy inputs intensity come from irrigation, fertilizers (N, P₂O₅, and K₂O) production and transport, machinery (includes tractors, harvesters, and threshers) manufacturing, and fuel consumption of machines. **b** Besides GHG emissions generated by the four energy inputs, soil emissions from N fertilizers use are considered. The useful life of machinery is assumed to be a 20-year average. The fuel used in the machinery mainly considers liquefied petroleum gas, motor gasoline, and gas-diesel oils. Other greenhouse gas emissions (e.g. CH₄ and N₂O) generated by energy inputs are converted to carbon dioxide equivalents (CO₂e), using global warming potential (GWP) values of the IPCC AR5 with no climate feedback (GWP-CH₄ = 28; GWP-N₂O = 265).

17. In L231, is the reduction due to a combination of drip and low-carbon electricity? If so, it is suggested to mention the exact combination of the scenarios because it is confusing to the readers as the paper deals with a number of different scenarios.

Revised manuscript: Line 275-285 in main text

Response: Sorry about the vague expression. According to the Reviewer’s suggestion, we have modified this vague expression and made it clear that the reduction was due to the combination of drip irrigation with low-carbon electricity. In addition, we also make it clear in the figure caption (Fig. 5).

Fig. 5 Feasibility of solutions to reduce CO₂ emissions of irrigation on a country-level scale. a Feasibility (%) of drip and low-carbon electricity. **b** Potential contributions (%) of drip and low-carbon electricity to energy-related CO₂ emissions reduction based on the feasibility analysis. The pie chart shows the contribution ratio of low-carbon electricity and drip to energy-related CO₂ emissions reduction. The values at the bottom of the pie chart represent the total contribution due to a combination of the two solutions. For a more detailed comparison of regional differences, we further divided the six continents into nine sub-regions.

18. The discussion section seems to be very lightly written given the effort that has been applied to the other sections (particularly the analysis and presentation) of the paper. There are many places where the message is not delivered clearly. For example, the authors argue that water saving technologies such as sprinkler and drip are more energy demanding. However, it is not clear from the discussion how a rational trade-off can be achieved in this regard.

Revised manuscript: Discussion section in main text

Response: According to the Reviewer's suggestion, in our revised discussion section, we discussed how to achieve a reasonable trade-off from the following four aspects: (1) Comparison of drip and sprinkler irrigation systems in terms of energy and CO₂ emissions reductions and water savings. And we give recommendations for priority deployment; (2) through a case study, we discuss the trade-off between water savings

and reductions of energy and CO₂ emissions for drip irrigation systems adoption; (3) we discussed how to achieve this trade-off through benefit-cost analysis; (4) we propose to incorporate this benefit-cost analysis into the trade-offs of water savings versus energy and carbon emissions reductions for countries across the world. The details are as follows:

“Drip and sprinkler represent two water-efficient irrigation systems. Still, our results show that sprinkler irrigation system has higher energy and CO₂ emissions intensity than surface irrigation and does not reduce energy use and CO₂ emissions of irrigation globally (Fig. 1b,d and Fig. 4). Therefore, priority should be given to drip irrigation system in the deployment of farm infrastructure. The exception is when switching from gravity irrigation to drip irrigation increases energy consumption and CO₂ emissions of irrigation, where there is a trade-off between water savings and reductions of energy and CO₂ emissions ^[1], as exemplified by Sudan and Ethiopia (Supplementary Fig. 5 and 15). In this case, a benefit-cost analysis should be incorporated into the trade-off. If carbon emissions are also considered as an investment cost, then the cost increase induced by the adoption of drip irrigation systems includes the initial capital investment of equipment with benefits throughout the life cycle, usually 15-25 years, the converted cost of energy input, and carbon taxes ^[2]. The economic returns from investments in drip irrigation technology include improvement in production, a shift in cropping rotation, and water and fertilizer savings ^{[2],[3]}. If the benefits are greater than the costs, the investment in drip irrigation systems can be treated as economically viable in countries like Sudan and Ethiopia. The benefit-cost analysis also applies to other countries around the world.”

References:

- [1] Mushtaq, S., Maraseni, T. N. & Reardon-Smith, K. Climate change and water security: Estimating the greenhouse gas costs of achieving water security through investments in modern irrigation technology. *Agric. Syst.* 117, 78-89 (2013).
- [2] Maraseni, T. N., Mushtaq, S. & Reardon-Smith, K. Climate change, water security and the need for integrated policy development: the case of on-farm infrastructure investment in the Australian irrigation sector. *Environ. Res. Lett.* 7, 34006-34012 (2012).
- [3] Maraseni, B. T. & Cockfield, G. Including the costs of water and greenhouse gas emissions in a reassessment of the profitability of irrigation. *Agric. Water Manage.* 103, 25-32 (2012).

19. Additionally, the argument that higher irrigation efficiency rarely reduces water consumption (L304-305) seems to be an understatement. Isn't it logical to want to increase the irrigation (command) area or cropping intensity if water is saved and can be used for other purposes?

Revised manuscript: Discussion section in main text

Response: Sorry about the illogical statement. In our revised discussion section, we discuss the benefits of drip irrigation adoption for improvement in production, a shift in cropping rotation, and water and fertilizer savings.

20. It is not clear how the authors arrived at different estimations such as the irrigation energy intensity values of 4 GJ/ha and energy required to produce and transport fertilizers (3 GJ/ha) and carbon emission intensity (415 kg CO₂/ha) (L266-275).

Revised manuscript: ----

Response: In our previous manuscript, the global average energy input (4 GJ/ha) and carbon emission (415 kg CO₂/ha) intensity of irrigation is based on the first section (Energy and CO₂ emissions intensity of irrigation) of results of this study. The energy input intensity (3 GJ/ha) of fertilizers produce and transport represents the global average values, which can be calculated based on country-level fertilizer use (kg/ha) derived from FAOSTAT and time-varying energy conversion factors (GJ/t) derived from Pellegrini et al. (2018). In our revised discussion, we focused on highlighting the high energy input and carbon emissions of irrigation, and weakened the discussion of the energy consumption and carbon emissions from fertilizers, machinery, and fuels.

References:

[1] Pellegrini P, Fernández R J. Crop intensification, land use, and on-farm energy-use efficiency during the worldwide spread of the green revolution[J]. Proceedings of the National Academy of Sciences, 2018, 115(10): 2335-2340.

21. Moreover, the values presented in L280-288 do not correspond with Figure 6.

Revised manuscript: Supplementary Table 8

Response: Sorry for the lack of clarity. In our revised manuscript, we make it clear that these values (the proportion of irrigation energy input/CO₂ emissions in total energy input/CO₂ emissions) correspond to Supplementary Table 8.

Supplementary Table 8. Summary of global energy and CO₂ emissions from farm operations during 2000-2010.

Stages of agriculture	Total Energy inputs (PJ per year)	Total CO ₂ emissions (Mt CO ₂ per year)
Fertilizers production and transport	4859	491
Fertilizers use	-	313
Machinery	1583	150
Fuel	3596	260
Irrigation	1896	216
Total	11934	1430

Note: Total energy inputs and CO₂ emissions estimations from other farm operations can be calculated as energy input intensity and CO₂ emissions intensity multiplied by total cropland area.

22. The authors mention “stringent low-carbon electricity targets” in L289-90. However, they fail to explain what such stringent targets are. Similarly, it is not clear how the authors got the figures presented in L292-296 that have been referred to Supplementary materials Fig 20a (which is not accessible). Moreover, these paragraphs (L266-296) are results and is strongly suggested to be moved to the results section.

Revised manuscript: ----

Response: Sorry for the loose statement. In our revised manuscript, we removed the undefined expression (stringent low-carbon electricity targets) and rewrote the discussion towards key points from our results. In addition, according to reviewer’s suggestion, we moved the analysis (Line 292-296 of the previous manuscript) of sustainable irrigation expansion in discussion into the results (Future energy consumption and CO₂ emissions, Fig. 2), and incorporated energy and CO₂ emissions from sustainable irrigation expansion in previous Supplementary Fig. 20 into new Figure 2 using a stacked bar graph. In the improved Figure 2, the values of energy consumption and CO₂ emissions from sustainable irrigation expansion are clear.

In addition, according to the Reviewer’s suggestion, we move the analysis (Line 266-288 of previous manuscript) of energy intensity and carbon emission intensity to the results section (**Irrigation contribution to on farm energy use**). Please see the response to comment (16).

Fig. 2 Global energy consumption and CO₂ emissions from irrigation. Country-level energy consumption and CO₂ emissions are based on pixel sum statistics. **a** Energy consumption (PJ per year) under 2000-2010 and sustainable irrigation expansion of 3 °C warmer climate. **b** CO₂ emissions (Mt CO₂ per year) from energy consumption under 2000-2010 and sustainable irrigation expansion of 3 °C warmer climate. We selected the top 20 countries with the highest energy consumption and CO₂ emissions. The upper right subgraphs represent a summary of energy consumption and CO₂ emissions by regions as well as globally.

23. There are instances in which the authors report using estimates dating back to the 2000s. It is understandable that these values are considered appropriate for use in the analysis from 2000-2011. However, the paper does not discuss at any point in the paper how these values have changed in the last decade and what impacts they could have on the energy use and emissions. Although a quantitative comparison could be difficult, it

would be extremely meaningful if the authors, based on their expertise, shed some light on this as we are almost one and a half decades from the analysis period of this paper where the energy and emission conditions have changed considerably.

Revised manuscript: Line 380-380 in main text; Supplementary Fig. 24

Response: We agree with reviewer's opinion. According to reviewer's suggestion, we added the discussion on the impact of energy consumption and CO₂ emissions due to the increase in irrigated area in the last decade. Based on our search of current irrigation water withdrawal datasets, we found that estimates of irrigation water withdrawal are still before 2010 ^[1]. Although FAO AQUASTAT provides country-level irrigation water withdrawal datasets for the most recent year, irrigation water withdrawal data is missing in some major irrigation-intensive countries such as India, China and Pakistan, which prevents our study from providing the most time-sensitive analysis of global energy consumption and CO₂ emissions of irrigation.

However, Puy et al. (2021) states that irrigation water withdrawals are driven by irrigated areas ^[2]. Therefore, we still have the opportunity to provide an updated understanding of global energy consumption and CO₂ emissions of irrigation after 2010, assuming all other conditions except irrigated area remain at 2000-2010. Based on energy and CO₂ emissions intensity per unit of irrigated area (Fig. 1 a,c of the main text) and irrigated area derived from FAO AQUASTAT in 2020, global energy consumption and CO₂ emissions of irrigation will continue to increase by about 14% in 2020 compared to 2000-2010, with significant increases in some countries of Africa (Supplementary Fig. 24). Although the assumptions are not reasonable, we provide conservative estimates of irrigation energy consumption and CO₂ emissions for the last decade.

Supplementary Fig. 24. Changes of irrigation energy consumption and CO₂ emissions in 2020 compared with 2000–2010. Country-level energy consumption and CO₂ emissions of irrigation in 2020 are calculated by multiplying energy and CO₂ emissions intensity per unit of irrigation area (Fig. 1 a,c) with irrigation area in 2020. The country-level irrigation area is from FAO AQUASTAT.

References:

- [1] McDermid S, Nocco M, Lawston-Parker P, et al. Irrigation in the Earth system[J]. Nature Reviews Earth & Environment, 2023: 1-19. (Supplementary Table 2)
- [2] Puy A, Borgonovo E, Lo Piano S, et al. Irrigated areas drive irrigation water withdrawals[J]. Nature communications, 2021, 12(1): 4525.

24. Moreover, the discussion needs to be directed towards some concrete points that can be derived from the results of this research.

Revised manuscript: Discussion section in main text

Response: We agree with reviewer's opinion. According to reviewer's suggestion, in our revised manuscript, we have rewritten the discussion. Details are as follows:

“Since the Green Revolution of the 1960s, global crop production has increased by 3.7 times due to the intensification, mechanization and modernization of agricultural systems. However, gains in yield have come at a considerable cost in terms of increased energy input and considerable environmental footprint. Irrigation development, as the concentrated embodiment of agricultural intensification, involved high energy consumption and led to reliance on fossil fuel, and CO₂ emissions (Fig. 2). In addition, energy input and carbon emissions from irrigation contribution significantly to total energy consumption and carbon emissions in agriculture (Fig. 6 and Supplementary Table 8).

High energy input and carbon footprint of irrigation in turn would potentially threaten the growth and stability of food production worldwide, especially in regions heavily dependent on fossil fuels. Furthermore, the high energy input of irrigation also increases the pressure on the energy supply system and competition for energy from other sectors. In Pakistan and Bangladesh, energy consumption of irrigation alone accounts for 4% of total energy supply in 2000-2010 (Supplementary Fig. 20a). With future sustainable irrigation expansion (Fig. 2), additional energy consumption of irrigation will add pressure on energy supply in African and European countries (Supplementary Fig. 20b).

We provide solutions for achieving low energy consumption and carbon emissions as well as highly efficient water use in irrigated agriculture (Fig. 4 and 5). Drip and sprinkler represent two water-efficient irrigation systems. Still, our results show that sprinkler irrigation system has higher energy and CO₂ emissions intensity than surface irrigation and does not reduce energy use and CO₂ emissions of irrigation globally (Fig. 1b,d and Fig. 4). Therefore, priority should be given to drip irrigation system in the deployment of farm infrastructure. The exception is when switching from gravity irrigation to drip irrigation increases energy consumption and CO₂ emissions of

irrigation, where there is a trade-off between water savings and reductions of energy and CO₂ emissions, as exemplified by Sudan and Ethiopia (Supplementary Fig. 5 and 15). In this case, a benefit-cost analysis should be incorporated into the trade-off. If carbon emissions are also considered as an investment cost, then the cost increase induced by the adoption of drip irrigation systems includes the initial capital investment of equipment with benefits throughout the life cycle, usually 15-25 years, the converted cost of energy input, and carbon taxes. The economic returns from investments in drip irrigation technology include improvement in production, a shift in cropping rotation, and water and fertilizer savings. If the benefits are greater than the costs, the investment in drip irrigation systems can be treated as economically viable in countries like Sudan and Ethiopia. The benefit-cost analysis also applies to countries around the world.

Remarkably, drip irrigation is not applicable to all crops (Supplementary Table 6), and its contribution to reducing global CO₂ emissions of irrigation is limited (Fig. 4b and Fig. 5b). Another solution in our study is to switch from energy-intensive diesel to efficient electric pumping (Fig. 1b) and use low-carbon electricity. Low-carbon electric pumps have a significant effect on reducing CO₂ emissions of irrigation (Fig. 4b and Fig. 5b), and have the same efficacy as drip irrigation in reducing energy consumption of irrigation (Fig. 4a). Low-carbon electricity brings substantial long-term benefits to the country, such as breaking away from fossil fuels dependence. However, from farmer's perspective, the reduction of CO₂ emissions is not a priority. An introduced carbon price may facilitate the adoption of low-carbon electricity, but it may also erode the relative profitability of irrigated crop production. Therefore, efforts should be put into the development of low-carbon electricity and cost reduction for countries worldwide in the future. On the other hand, global energy consumption and CO₂ emissions from irrigation are dominated by groundwater pumping (Fig. 3). In this case, it is recommended to give priority to the use of surface water and shallow groundwater for irrigation. Meanwhile, the management of groundwater resources should be strengthened to prevent the decline in groundwater levels from offsetting the energy efficiency gains from the adoption of drip irrigation systems.

Currently, spatially-explicit estimates of global irrigation water withdrawals for irrigation are still before 2010, which prevents our study from providing the most time-sensitive analysis of global energy consumption and CO₂ emissions of irrigation. However, previous studies have shown that irrigation water withdrawal is driven by irrigated area. Thus, we still can provide an updated understanding of global energy consumption and CO₂ emissions of irrigation after 2010, assuming all other conditions except irrigated areas remain as in 2000-2010. Based on energy and CO₂ emissions intensity per unit of irrigated area (Fig. 1 a,c) and country-specific irrigated area derived from FAO AQUASTAT in 2020, global energy consumption and CO₂ emissions

of irrigation increased by about 14% in 2020 compared to 2000–2010, with significant increases in some African countries (Supplementary Fig. 24).

Our study sheds light on the previously uncharted territory of global energy consumption and carbon emissions associated with irrigation. This research not only provides a comprehensive understanding of global direct energy use and associated CO₂ emissions from irrigation but also charts a path forward aiming for less water, energy, and CO₂ emissions in irrigated agriculture. Furthermore, previous work showed biophysical limits to irrigation showing where water is locally available to meet crop water demand. However, irrigation is not only influenced by water availability but also socio-economic factors, a phenomenon known as agricultural economic water scarcity. Mapping at high resolution the energy and carbon emissions from irrigation allows us to understand where energy will be a barrier to irrigation. Our study provides detailed information on energy usage from irrigation and can inform on the feasibility of irrigation to increase adaptive capacity in the agricultural sector.”

25. Although the paper is relatively well-structured, the language needs considerable refinement. There are incomplete/grammatically incorrect sentences for example L486-489. There are some sentences which are not clear, for example L234-235. There are a number of repetitions among different sections of the paper, for example, L234-245, L321-325 and L333-336.

Revised manuscript: Line 405-409 in main text

Response: Sorry for the language flaw. First, the co-authors of the manuscript conducted a thorough and careful examination of the language, logic, and expression in the manuscript. Second, we have adopted a paid English language editing service from Editage to improve the English writing.

We have deleted the unclear sentences in lines 234-235 of the discussion and rewritten the discussion. In addition, we modified and merged the repeated statements in lines 321-325 and 333-336 of the methods section.

26. The figures and maps presented at the end of the pdf file are not numbered and their caption is missing; some figures are repeated a number of times. In many instances, the units are missing and their references in the main text is not clear. It is suggested that the authors take utmost care in sorting these issues.

Revised manuscript: ----

Response: Sorry for our mistake, in our revised manuscript, we add captions as well as units for each figure to make it easier to understand (e.g. Supplementary Fig.8). Furthermore, we make explicit the references to the graph in the main text. Since the figures was merged into the pdf file after uploading, the picture number information is

missing. So, we provide a separate pdf figures file with the full information of figures for better review.

Supplementary Fig. 8. Global energy consumption and CO₂ emissions from surface water pumping and delivery. **a** Energy consumption. **b** CO₂ emissions.

27. Some highly relevant papers are missing. Some of them are given below.

- Energy and water tradeoffs in enhancing food security: A selective international assessment, *Energy policy*, 2009; 37 (9), 3635-3644
- Climate change and water security: estimating the greenhouse gas costs of achieving water security through investments in modern irrigation technology, 2013; *Agricultural Systems* 117, 78-89
- Carbon smart agriculture: An integrated regional approach offers significant potential to increase profit and resource use efficiency, and reduce emissions; 2021, *Journal of Cleaner Production* 282, 124555
- Climate change, water security and the need for integrated policy development: the case of on-farm infrastructure investment in the Australian irrigation sector, 2013, *Environmental Research Letters* 7 (3), 034006

Revised manuscript: References in main text [36], [37], [38], [39], [40], [41]

Response: We thank you for your constructive suggestions as well as for providing very valuable references, which will help us improve the discussion section of this study. In addition, two other very valuable references ^{[1], [2]} also helped us improve the discussion. In our revised manuscript, we cited the following six references.

Again, special thanks to you for your good comments and suggestions.

References:

- [1] Maraseni T N, Cockfield G. Including the costs of water and greenhouse gas emissions in a reassessment of the profitability of irrigation[J]. *Agricultural water management*, 2012, 103: 25-32.
- [2] Bundschuh, Jochen, et al., eds. "Geothermal, wind and solar energy applications in agriculture and aquaculture." (2017).
- [3] Mushtaq S, Maraseni T N, Reardon-Smith K. Climate change and water security: estimating the greenhouse gas costs of achieving water security through investments in modern irrigation technology[J]. *Agricultural Systems*, 2013, 117: 78-89.
- [4] Mushtaq S, Maraseni T N, Maroulis J, et al. Energy and water tradeoffs in enhancing food security: A selective international assessment[J]. *Energy policy*, 2009, 37(9): 3635-3644.
- [5] Maraseni T N, Mushtaq S, Reardon-Smith K. Climate change, water security and the need for integrated policy development: the case of on-farm infrastructure investment in the Australian irrigation sector[J]. *Environmental Research Letters*, 2012, 7(3): 034006.
- [6] Maraseni T, An-Vo D A, Mushtaq S, et al. Carbon smart agriculture: An integrated regional approach offers significant potential to increase profit and resource use efficiency, and reduce emissions[J]. *Journal of Cleaner Production*, 2021, 282: 124555.

Reviewer #3:

1. I enjoyed reading this article very much. The authors have focused on an important issue, energy and carbon use for irrigation globally, and crafted some beautiful graphics to illustrate their analytical findings. Overall, I thought the manuscript was well written, the scenarios were compelling and interesting, and I applaud the authors for the steps they took to create global estimates with this physics-based method.

I am going to identify myself in this review, because I will be citing some of my group's work. My name is Anthony Kendall, I am a hydrogeology professor at Michigan State University. In 2020, my master's student Ben McCarthy, published what I believe to be the first large-scale physics-based analysis of energy and carbon emissions from irrigation (McCarthy et al. 2020), focused on the US High Plains Aquifer in Kansas. My expertise in this subject is derived from the years I spent working on this analysis with Ben, in direct coordination with my co-authors including Annick Anctil, a professor of environmental engineering specializing in life cycle analyses.

Before publication, I believe that the authors should enhance their methodology as described below. Without this, their methods are capable of making global maps but can be substantially incorrect (as is in the case in the US). I understand the need to simplify methodology at the global scale, but their conclusions really depend entirely on their methods being accurate. Furthermore, it's not clear that the errors introduced by their methodological oversights are random. In particular, groundwater energy use would appear to understated--in some cases significantly.

I have three primary issues, and several smaller ones, that I believe should be addressed before further consideration of this manuscript occurs.

Response: Thank you very much for taking your valuable time to review this manuscript. We really appreciate all your valuable comments and suggestions. Your constructive comments have greatly helped to improve the reliability of the methodology, and contributed additional novel content (CO₂ emissions from groundwater due to degassing). In particular, your valuable suggestions contribute to quantifying global groundwater drawdown depth in a simplified way and a good constraint on total efficiencies of pump in this study, which substantially helped to improve the quality of the manuscript. Next, we make a point-by-point response.

1. ## Calculation of Direct Energy Requirements

First, the authors' calculation of energy requirements for groundwater pumping omits several important factors that can play an important role in accurately computing energy

use. The components of the energy required to lift water from the inside of the well bore to the irrigation nozzles/drip outlets is acutally calculated as:

$E_{tot} = \rho * g * V * h_{tot} * \eta_p$; where ρ = the density of water, g = the gravitational constant, V = the volume pumped, η_p = the combined pump and prime mover efficiency, and h_{tot} = the total lift height, which is defined as:

$h_{tot} = h_{lift} + h_{press} + h_{loss}$; where h_{lift} = the lift height, h_{press} = the pressurization in the pipe for each system, and h_{loss} = primarily pump losses.

So far, the authors and I agree on these definitions, though our equations are somewhat different (due to units in the total direct energy, mostly). Where my issue arises is in the h_{lift} component. Because the height that water needs to be lifted isn't just the depth to the average regional water table, but rather the sum of the depth to the regional water table, the additional depth of the drawdown cone formed around each well as the pumping season progresses, and the additional depth of water inside the well bore caused by friction in the well screen and well packing material. Thus,

$h_{lift} = h_{wt} + h_{dd} + h_{we}$; where h_{wt} = the water table lift, h_{dd} = the depth of the drawdown cone (time-varying), and h_{we} = depth of water in the well below that of the area immediately outside it.

The authors' omission of the h_{dd} and h_{we} terms can be highly significant, particularly for large irrigated farms with high pumping flow rates. In particular, for areas with shallow water tables, these terms can dominate the h_{lift} . Calculating these two terms is not incredibly difficult, at least if we make simplifying assumptions. See McCarthy et al. (2020) and its supplementary information for details.

Revised manuscript: Supplementary Methods section 1.1

Response: We agree with the reviewer's opinion. In previous version, although we tried to consider the depth of the drawdown cone through an empirical formula ^[1], it is obvious that the reviewer's proposed method is more reliable and considered depth of water in the well below that of the area immediately outside it. Therefore, we calculated the depth of the drawdown cone (L_{CD}) and the additional lift due drawdown within the well as a result of frictional losses in and within the immediate vicinity of the well (L_{WD}) according to the method used by McCarthy et al. (2020) (Section 1.8 of supplementary information and materials in HydroShare). The calculation formula is as follows:

$$L_{cd} = \frac{Q}{4\pi T} \left[-0.5772 - \ln\left(\frac{r^2 S}{4Tt}\right) \right] \quad (0.1)$$

where L_{cd} is the depth of drawdown cone; Q is the pump rate (m^3/day); T is the transmissivity, which can be obtained by multiplying hydraulic conductivity (m/day) and the saturate thickness (m) of aquifer. S is the specific yield (dimensionless); r is the well radius (m), which remains consistent with the McCarthy et al. (2020); t is the time pumped (day); Assuming a well efficiency of 50%, which remains consistent with the McCarthy et al. (2020). Therefore, L_{wd} can be obtained by multiplying L_{cd} by 0.5.

Since some parameters in the formula are difficult to obtain and the calculation of these parameters is extremely complex and computationally intensive on a global scale, we have made reasonable assumptions in the calculation. In the following, key parameters will be described in detail.

(1) Pump rate (m^3/day)

As for pump rate, we used the regression parameters (derived from McCarthy et al. 2020) of pump rate and annual water use for flood, trickle-drip, and center pivot irrigations techniques. We assume that these three irrigation techniques represent typical surface irrigation, drip irrigation, and sprinkler irrigation systems.

(2) Hydraulic conductivity (K , m/day) and specific yield (S_y)

Hydraulic conductivity and specific yield are the two principal hydraulic characteristics that control groundwater flow in a water-table aquifer. Both hydraulic conductivity and specific yield depend on the character of the sediments that comprise the aquifer; their values can be expected to vary both horizontally and vertically according to the variation in sediment types. According to McCarthy et al. (2020), the number of pump wells with hydraulic conductivity of 22.86 and 45.72 accounted for 79%, and the number of pump wells with specific yields of 0.125 and 0.175 accounted for 72%. This indirectly reflects that good hydraulic conductivity and specific yield are important conditions for drilling selection.

At present, although Gleeson et al. (2014) provided the only available global near-surface lithological permeability datasets, we found that the datasets only represented the hydraulic conductivity of near-surface lithology, and could not truly reflect the hydraulic conductivity of saturated aquifers. And the values differ by several hundred times from that of aquifer conductivity reported in general literature.

Therefore, according to McCarthy et al. (2020), we assume an average hydraulic conductivity of 34.29 and a specific yield of 0.15 for a global analysis.

(3) Saturate thickness (m) of aquifer

Given the availability of data, here we chose depth to bedrock datasets at a resolution of 5 arc-minutes derived from Wei et al. (2017) to reflect the thickness of the aquifer.

(4) Pump time (day)

We estimated the pump time based on crop calendar datasets at a resolution of 5 arc-minutes (start and end months of crop plant) derived from Portmann et al. (2010), and length of each growing stage (includes initial, development, middle and late stages) of crops (Table 2 of Siebert et al. (2008)). Specifically, we first calculate the number of days of the entire growing season for each crop based on the start and end months. Then, we assume that the first three stages (initial, development, middle) of crop growth are the periods when the crop needs the most water, and further calculate the average pump days for each grid based on a reasonable irrigated frequency every three days.

Results of global groundwater drawdown due to irrigation:

The most significant groundwater drawdowns mainly distributed in global large irrigated areas (e.g., Central of the United States, northern of India, Pakistan, and the North China Plain). The global average drawdown is 6.99 m for surface irrigation system, 6.22 m for sprinkler irrigation system, and 4.43 m for drip irrigation system.

Figure.1 Geospatial distribution of global groundwater drawdown depth due to groundwater pumping using different irrigation techniques. **a** Surface irrigation system. **b** Sprinkler irrigation. **c** Drip irrigation

References:

- [1] Wang J, Rothausen S G S A, Conway D, et al. China' s water – energy nexus: greenhouse-gas emissions from groundwater use for agriculture[J]. Environmental Research Letters, 2012, 7(1): 014035.
- [2] McCarthy B, Anex R, Wang Y, et al. Trends in water use, energy consumption, and carbon emissions from irrigation: role of shifting technologies and energy sources[J]. Environmental Science & Technology, 2020, 54(23): 15329-15337.
- [3] Gleeson T, Moosdorf N, Hartmann J, et al. A glimpse beneath earth's surface: GLobal HYdrogeology MaPS (GLHYMPS) of permeability and porosity[J]. Geophysical Research Letters,

2014, 41(11): 3891-3898.

[4] Shangguan W, Hengl T, Mendes de Jesus J, et al. Mapping the global depth to bedrock for land surface modeling[J]. *Journal of Advances in Modeling Earth Systems*, 2017, 9(1): 65-88.

[5] Portmann F T, Siebert S, Döll P. MIRCA2000—Global monthly irrigated and rainfed crop areas around the year 2000: A new high-resolution data set for agricultural and hydrological modeling[J]. *Global biogeochemical cycles*, 2010, 24(1).

[6] Siebert S. The Global Crop Water Model (GCWM): Documentation and first results for irrigation crops[J]. *Frankfurt Hydrology Paper*, 2008, 7: 1-42.

2. ### "Pump" vs. "Prime Mover" efficiency

One of the reasons that the authors encountered such a wide range of so-called "pump efficiency" estimates is due to the fact that what we might colloquially call "pump efficiency" is actually two things: true pump efficiency, including impeller, driveshaft, and other losses, and; prime mover efficiency, or the efficiency that the electric, diesel, natural gas, or other motor converts its energy source to mechanical movement. Pump efficiencies are usually quite high, on the order of 70% or so for a well-maintained system. Prime mover efficiencies can be much lower, on the order of 25-30% for diesel and natural gas systems, and 80% or so for small electric motors (running on single-phase AC), to 90% or more for large electric motors (using three-phase AC). Thus, total efficiencies are approximately 20% for diesel-driven pumps, and 56% for small electric/63% for large electric pumps. Clarity on this point would help bolster the credibility of the authors' calculations.

Revised manuscript: Supplementary Methods section 1.2

Response: We agree with the reviewer's opinion. According to the Reviewer's suggestion, we use conservative pump efficiency of 70% for global analysis. In addition, we assume an electric motor efficiency of 80%, a diesel engine efficiency of 30%, and a natural gas engine efficiency of 25%. The adoption of pump and prime mover efficiencies is illustrated in Supplementary Method Section 1.2 of our revised manuscript.

4. ### Lack of natural gas for the US

Based on the Supplementary Table 3, the US has at least 36% of its pumping provided by sources other than diesel- or electrically-driven pumps. Their citation for this is also woefully out-of-date, being nearly 40 years old. The US government publishes statistics on this, at the county level, every five years through its US Department of Agriculture Irrigation and Water Management Surveys. Now I realize that the US is just one country, but it is the second-largest consumer of energy and emitter of carbon for irrigation, according to these authors. It is also the place where the most study has gone into this issue (more on that below), and the best statistical data are available to validate their approach.

I am also wondering if the authors took the proportion of diesel and electric pumps reported in their table S3 and divided by the sum, to scale the pump proportions to 100%. I assume so, but this is not explicitly stated that I could tell.

Revised manuscript: Supplementary Methods section 1.4

Response: We agree with the reviewer's opinion. According to the Reviewer's suggestion, we use county-level pumps data (in 2003) for the US. The county-level pumps data (include pumps powered by electricity, diesel, and natural gas) was derived from the US government publishes statistics (Table 20-in reference) ^[1]. In addition, we have updated the attached citation for irrigation pumps.

References:

[1] USDA NASS (United States Department of Agriculture, National Agricultural Statistics Service). Farm and Ranch Irrigation Survey, 2003[J]. 2003.

5. # Lack of direct emissions of CO₂ from groundwater due to degassing

When groundwater is pumped, it may be supersaturated in carbonate relative to atmospheric pressure. The excess carbonate will then degas, producing direct CO₂ emissions at the irrigated field. This is explained in Wood and Hyndman (2017). McCarthy et al. (2020) show that direct CO₂ emissions from groundwater irrigation can be 30% or more of total CO₂ emissions in regions where groundwater is being actively depleted (as is true in most of the groundwater-irrigated areas worldwide). Inclusion of this process would not be excessively difficult, and simplifying assumptions could be used to produce global estimates.

Revised manuscript: The second part of the results; Methods section in main text; Supplementary Fig. 6;

Response: We agree with the reviewer's opinion. According to the Reviewer's suggestion, in our revised version, we have carried out an analysis of direct CO₂ emissions from groundwater degassing caused by irrigation. The calculation formula is as follows:

$$C_{GD} = V_{GD} * C_{GW} \quad (0.2)$$

Where C_{GD} is CO₂ emissions from groundwater degassing caused by irrigation; V_{GD} is groundwater depletion volume due to irrigation (m³); C_{GW} is CO₂ concentration in groundwater.

(1) Groundwater depletion due to irrigation

We used global average annual groundwater use datasets during 2000-2009 with a resolution of 0.5° derived from Döll et al. (2014). The datasets are based on WaterGAP 2.2 model combined with local well observation or GRACE satellite observation, which

can largely reduce uncertainty inherent in flux-based method [2]. In addition, in our analysis, groundwater depletion is caused by groundwater extraction rather than climate change (anthropogenic influence: GWD IRR70_S minus GWD NOUSE_S, Table 6 of Döll). The global groundwater depletion caused by groundwater extraction is 113 km³/yr during 2000 – 2009.

However, Döll (2014) did not estimate groundwater use due to irrigation pumping separately. Therefore, we estimation groundwater use of irrigation by multiplying the proportion of groundwater use from irrigation to total groundwater extraction by total groundwater depletion. According to Döll et al. (2014) (Table 2), groundwater use comes mainly from irrigation, domestic, and manufacturing sectors. Groundwater use of domestic accounts for 36% of total domestic water use, and groundwater use of manufacturing accounts for 26% of total manufacturing water use. The total water use of domestic and manufacturing datasets can be obtained from Huang et al. (2018).

(2) CO₂ concentration in groundwater

According to Desimone et al. (2009) and Wood and Hyndman (2017), the HCO₃⁻ concentration of the 25 percentiles of the 2003 groundwater samples is 95 mg/L and the 75 percentile is 293 mg/L in the United States. We assume that bicarbonate content of groundwater on earth is likely like to that measured in the United States (95 mg/L to 293 mg/L). Then, the CO₂ concentration (mg/L) can be calculated by multiplying the molecular mass ratio of HCO₃⁻ and CO₂ with the bicarbonate concentration. The calculation formula is as follows:

$$CO_2 \text{Concentration} = \frac{1}{2} HCO_3^- \times \frac{44}{61} \quad (0.3)$$

Finally, the concentration of CO₂ in groundwater ranges from 34.26 mg/L to 105.67 mg/L.

CO₂ emissions from groundwater degassing caused by irrigation:

We estimate that global average annual CO₂ emissions from groundwater degassing caused by irrigation are 6.41 Mt CO₂ (3.14~9.67 Mt CO₂) (Supplementary Fig. 6a). India has the largest CO₂ emissions of 2.92 Mt CO₂, followed by the United States (1.43 Mt CO₂), Iran (0.51 Mt CO₂), China (0.36 Mt CO₂), Saudi Arabia (0.33 Mt CO₂), which account for 46%, 22%, 8%, 6%, and 5% of the total CO₂ emissions from groundwater caused by irrigation, respectively. The values of CO₂ emissions from groundwater degassing (only irrigation) in the United States estimated in this study are relatively consistent with the results (1.7 Mt CO₂ per year) estimated by Wood and Hyndman (2017). Notably, in major irrigation-intensive regions of the United States, India, Pakistan, Iran, and Saudi Arabia, CO₂ emissions from groundwater degassing

account for more than 20% of total CO₂ emissions from irrigation (Supplementary Fig. 6b).

Supplementary Fig. 6. Geospatial distribution of CO₂ emissions from groundwater degassing from irrigation in 2000-2010. **a** CO₂ emissions from groundwater degassing. These CO₂ emissions are from the CO₂ molecules absorbed at high pressure in water molecules underground, which are released in the atmosphere once groundwater is pumped at atmospheric pressure. These CO₂ emissions are non-energy related emissions. **b** The share of CO₂ emissions from groundwater degassing relative to total energy related CO₂ emissions.

References:

- [1] Döll P, Müller Schmied H, Schuh C, et al. Global - scale assessment of groundwater depletion and related groundwater abstractions: Combining hydrological modeling with information from well observations and GRACE satellites[J]. *Water Resources Research*, 2014, 50(7): 5698-5720.
- [2] Wada Y. Modeling groundwater depletion at regional and global scales: Present state and future prospects[J]. *Surveys in Geophysics*, 2016, 37(2): 419-451.
- [3] Huang Z, Hejazi M, Li X, et al. Reconstruction of global gridded monthly sectoral water withdrawals for 1971–2010 and analysis of their spatiotemporal patterns[J]. *Hydrology and Earth*

System Sciences, 2018, 22(4): 2117-2133.

[4] DeSimone, L. A. (2009). Quality of water from domestic wells in principal aquifers of the United States, 1991–2004. US Geological Survey, 139.

[5] Wood W W, Hyndman D W. Groundwater depletion: A significant unreported source of atmospheric carbon dioxide[J]. Earth's Future, 2017, 5(11): 1133-1135.

6. # Disagreement of energy use estimates in the US

In their Supplement, section 2.1, they compare their estimates to those from Liu et al. 2016, and find a 0.83 R² fit between available countries. However, this is entirely driven by their general agreement for India--pulling this statistic much higher. In particular, the second highest energy user, the US, is dramatically overestimated in their study. This would be acceptable if not for the fact that there are other estimates of energy use for the US. In 2022, Sowby and Dicaldo (2022) published an energy estimate for the US that was 219 PJ/yr. Here, the authors estimate 418 PJ/yr. The source the authors' compare to, Liu et al., in their Supplementary Table 2, estimate this total at 141 PJ/yr. Sowby and Dicaldo point to other independent estimates for the US Department of Energy that put the total energy use squarely within the range of 200 PJ/yr, not 400+

Again, the US is just one country, but it is: 1) the second-largest consumer of energy for irrigation, and 2) the one with the best publication record and energy statistics. While it is an outlier in several respects, including the degree of industrialization of its irrigated agricultural sector, and the size of farms irrigated, getting their methods right in that country seems very important. Note that Pakistan also seems to be substantially overestimated.

Also, presenting this validation figure in a log scale would be much more informative across the very wide range of energy consumption.

Revised manuscript: Supplementary Discussion section 2.1; Supplementary Fig. 23; Supplementary Table 9

Response: As suggested by the reviewer, after methodological and data improvements (as in response to comments (2), (3), (4) above), we re-estimated energy consumption from irrigation. In addition, other results from regional statistical publication were used to validate the results of this study.

“In this study, global energy consumption from irrigation was 1896 PJ. According to Liu et al., 2016, the energy consumption from agricultural water source and conveyance was 2433 PJ (Supplementary Table 9), which was based on irrigation water withdrawal and energy intensity values for each water process and source. From the comparison of energy consumption on the national scale, we noted a significant correlation ($R^2 = 0.68$) between them (Supplementary Fig. 23). However, there were differences in some countries.

In the United States and Pakistan, the energy consumption estimated by this study is 30%-40% higher than the results estimated by Liu et al., 2016 (Supplementary Table

9). The energy consumption of India estimated in this study was 78% lower than that estimated by Liu et al., 2016. However, the results from other studies on energy consumption of irrigation in India, Pakistan, and the United States were consistent with the results of this study (Supplementary Table 9).

Accordingly, in India and China, the energy-related CO₂ emissions estimated in our study were 70 Mt CO₂ and 35 Mt CO₂, which are close to the estimation of 59-92 Mt CO₂ in 2009 by Shah et al., 2009 and 34-47 Mt CO₂ in 2010 by Zou et al., 2015.”

Therefore, the estimation of energy consumption from irrigation conducted by this study is more reliable.

Supplementary Table 9. Comparison of energy consumption and CO₂ emissions from irrigation between this study and previous studies.

Energy consumption (PJ)					CO ₂ emissions (Mt CO ₂)		Source
Global	The United States	China	India	Pakistan	China	India	
2433	141	261	953	75			Liu et al. 2016
	219						Sowby and Dicaldo. 2022
				103			Siyal and Gerbens-Leenes. 2022
			439				Can et al. 2009
						59-92	Shah et al. 2009
					34-47		Zou et al. 2015
1896	205	299	535	135	35	70	This study

Supplementary Fig. 23. Country-level comparison of energy consumption from irrigation between this study and Liu et al. (2016).

References:

- [1] Liu, Y. et al. Global and regional evaluation of energy for water. *Environ. Sci. Technol.* 50, 9736-9745 (2016).
- [2] Sowby R B, Dicaldo E. The energy footprint of US irrigation: A first estimate from open data[J]. *Energy Nexus*, 2022, 6: 100066.
- [3] Siyal A W, Gerbens-Leenes P W. The water–energy nexus in irrigated agriculture in South Asia: Critical hotspots of irrigation water use, related energy application, and greenhouse gas emissions for wheat, rice, sugarcane, and cotton in Pakistan[J]. *Frontiers in Water*, 2022, 4: 941722.
- [4] Can, D L R D., McNeil M, Sathaye J. India energy outlook: end use demand in India to 2020. Lawrence Berkeley National Lab. (LBNL), Berkeley, CA (United States), 2009.
<https://eta-publications.lbl.gov/sites/default/files/lbnl-1751e.pdf>
- [5] Shah, T. Climate change and groundwater: India's opportunities for mitigation and adaptation. *Environ. Res. Lett.* 4, 35005 (2009).
- [6] Zou, X. et al. Greenhouse gas emissions from agricultural irrigation in China. *Mitig. Adapt. Strateg. Glob. Chang.* 20, 295-315 (2015).

7. # Smaller Issues:

Feasibility of drip irrigation for soybean

At least in the western hemisphere, soybean is grown on massive fields that are not feasible (at least in the near future) to use drip irrigation. The citation the authors use in their Supplementary Table 5 points to a source that has a table stating that Soybean is feasible to use with drip. However, that table also lists maize as unsuitable for sprinkler irrigation, when we know very well that millions of ha of maize are irrigated with pivot and linear move sprinkler systems worldwide.

Revised manuscript: Supplementary Table 6; Supplementary Fig. 19a

Response: We agree with the reviewer's opinion. According to the Reviewer's suggestion, in our revised manuscript, we corrected the error and recalculated the feasibility of drip irrigation.

8. ## Missing References

Here are the three papers I have referenced above. I have no doubt there are others as well. A more thorough literature search would certainly turn up additional contributions.

- McCarthy et al. 2020, Trends in Water Use, Energy Consumption, and Carbon Emissions from Irrigation: Role of Shifting Technologies and Energy Sources
- Sowby and Dicaldo, 2022, The energy footprint of U.S. irrigation: A first estimate from open data
- Wood and Hyndman, 2017, Groundwater Depletion: A Significant Unreported Source of Atmospheric Carbon Dioxide

Revised manuscript: References in main text [24], [29], [42], [43]

Response: We thank you for your constructive suggestions as well as for providing very valuable references, which will help us improve our manuscript. In addition, another valuable references ^[1] was also cited in our discussion. In our revised manuscript, we cited the following four references.

Again, special thanks to you for your good comments and suggestions.

References:

[1] Smidt S J, Haacker E M K, Kendall A D, et al. Complex water management in modern agriculture: Trends in the water-energy-food nexus over the High Plains Aquifer[J]. *Science of the Total Environment*, 2016, 566: 988-1001.

[2] McCarthy B, Anex R, Wang Y, et al. Trends in water use, energy consumption, and carbon emissions from irrigation: role of shifting technologies and energy sources[J]. *Environmental Science & Technology*, 2020, 54(23): 15329-15337.

[3] Sowby R B, Dicataldo E. The energy footprint of US irrigation: A first estimate from open data[J]. *Energy Nexus*, 2022, 6: 100066.

[4] Wood W W, Hyndman D W. Groundwater depletion: A significant unreported source of atmospheric carbon dioxide[J]. *Earth's Future*, 2017, 5(11): 1133-1135.

9. ## Over-complexity of primary manuscript graphics

As I said above, the manuscript graphics are well crafted, and striking. However, each one is a unique graphical theme, requiring quite a bit of time to parse. Please try to simplify. I can tell that the influence of the IPCC graphics is strong here, but papers are not IPCC reports. The simpler the graphics, the clearer the story, and the higher the impact a paper can have.

Revised manuscript: ----

Response: We agree with the reviewer's opinion. According to the Reviewer's suggestion, in our revised manuscript, we have simplified some complex graphics, see responses to comments (10) and (11).

10. A few examples:

- In Figure 3, having the country in the Sankey diagram twice is quite difficult to use. Their supplemental figure 12 is a much nicer Sankey diagram, actually.

- In Figure 4, there are so many adornments that seem unnecessary. The arrow seems to point in the direction of the bar, which is not needed. The triangles are present in some bars but not all, and sometimes seem identical to the error bars, other times different.

- Figure 5 has a whole collection of themes (radar, pie, bar) and colors.

- I still haven't figured out the mysterious color wheel in each panel of Figure 6.

Revised manuscript: Figures in main text 1-6

Response: We agree with the reviewer’s opinion. According to the Reviewer’s suggestion, we have improved Figures. 3, 4, 5 and 6.

(1) Figure 3

We use a simplified Sankey diagram as suggested by reviewer. Since the energy and CO₂ emissions from natural gas pumping are mainly in the United States, and the natural gas pumps accounts for only 9% of the total pumps (electric, diesel and natural gas pumps) [1]. In addition, we found that natural gas pumps are mainly distributed in Kansas, Nebraska and Texas [1]. Therefore, the analysis of natural gas pumping is not shown in the figure. The improved figure is shown below:

Fig. 3 Sankey diagram of the distribution of energy consumption and CO₂ emissions from irrigation. a Energy consumption (PJ per year) during 2000–2010. **b** CO₂ emissions from embodied energy consumption (Mt CO₂ per year). In the figure, the total energy consumption and CO₂ emissions do not include those (values in parentheses) from natural gas pumping. Geospatial distribution maps are provided in Supplementary Figs. 8-14.

References:

[1] USDA NASS (United States Department of Agriculture, National Agricultural Statistics Service). Farm and Ranch Irrigation Survey, 2003[J]. 2003.

(2) Figure 4

In figure 4, we removed the unnecessary adornments as suggested by reviewer. In addition, in the electric pumping scenario, we take into account hydropower and the projected electricity mix in 2050 as suggested by another reviewer. The improved figure is shown below:

Fig. 4 Mitigation potentials of energy use and CO₂ emissions under different scenarios. **a** The sprinkler scenario and drip scenario have different water-use intensities. For the electric pumping scenario, the energy consumption of irrigation does not change with the CO₂ intensity of electricity. **b** In sprinkler and drip scenarios, CO₂ intensity of energy (electricity during 2000–2010) remains unchanged. In electric pumping scenarios, electricity comes from current electricity mix, solar, wind, nuclear, hydropower, or a mix of the four low-carbon electricity according to IEA net-zero by 2050 projections. The baseline scenario reflects energy consumption and energy-related CO₂ emissions during 2000–2010. The current electricity mix represents the electricity production structure during 2000–2010, where the electricity comes from fossil fuels, nuclear, hydro, geothermal, solar, wind, tide, wave, ocean and biofuels. Geospatial distribution maps are provided in Supplementary Figs. 15-17.

(3) Figure 5

In Figure 5, we simplified the graph and used only two themes and colors. The improved figure is shown below:

Fig. 5 Feasibility of solutions to reduce CO₂ emissions of irrigation on a country-level scale. a Feasibility (%) of drip and low-carbon electricity. **b** Potential contribution (%) of drip and low-carbon electricity to energy-related CO₂ emissions reduction based on the feasibility analysis. The pie chart shows the contribution ratio of low-carbon electricity and drip to energy-related CO₂ emissions reduction. The values at the bottom of the pie chart represent the total contribution due to a combination of the two solutions. For a more detailed comparison of regional differences, we further divided the six continents into nine sub-regions (See Supplementary Data for the rationale of the classification).

(4) Figure 6

To avoid causing misunderstanding, we removed the analysis of the six continents and retained the more detailed analysis of the nine sub-regions. The improved figure is shown below:

Fig. 6 Contribution of irrigation energy input intensity to total energy input intensity and comparison of CO₂ emissions intensity generated by energy inputs intensity during 2000–2010. **a** Energy inputs intensity come from irrigation, fertilizers (N, P₂O₅, and K₂O) production and transport, machinery (includes tractors, harvesters, and threshers) manufacturing, and fuel consumption of machines. **b** Besides GHG emissions generated by the four energy inputs, soil emissions from N fertilizers use are considered. The useful life of machinery is assumed to be a 20-year average. The fuel used in the machinery mainly considers liquefied petroleum gas, motor gasoline, and gas-diesel oils. Other greenhouse gas emissions (e.g. CH₄ and N₂O) generated by energy inputs are converted to carbon dioxide equivalents (CO₂e), using global warming potential (GWP) values of the IPCC AR5 with no climate feedback (GWP-CH₄ = 28; GWP-N₂O = 265).

11. ## Other small graphical issues

- In Figure 1, the dashed line is a 1:1 line, but there is no basis for there to be a 1:1 fit between water intensity and energy/CO₂ intensity. Instead, a linear fit would then help the reader distinguish between countries above or below median energy/CO₂ intensity
- In Figure 2b, the orange bars fade to dark at low values, while there is either no fade, or it is reversed and barely visible in 2a

Revised manuscript: ----

Response: We agree with the reviewer’s opinion. According to the Reviewer’s suggestion, we have improved Figures .1a,c and 2.

(1) Figure 1

We attempt to replace the 1:1 line with a linear fit line. However, we found that the linear fit line did not perform well in distinguishing between countries above or below median energy intensity of water use. For example, when a linear fit line is used, European countries will lie below the linear fit line. In Figure .1a,c, although European countries have lower energy and CO₂ emissions intensity per unit of irrigation area, energy and CO₂ emissions intensity of water use (the ratio of the vertical and horizontal values) are the highest. Therefore, as an alternative, we use a line whose slope is the median of energy and CO₂ emissions intensity of water use to distinguish countries above or below the median (Fig. 1a,c).

Fig. 1 Energy and CO_2 emissions intensity as a function of water-use intensity for 159 countries. The figure shows the comparison of energy and CO_2 emissions intensity of different irrigation and pumping systems. **a** Energy intensity is expressed as the ratio of the energy consumed by irrigation to the irrigated area in a country (GJ/ha). **b** The energy intensity of each irrigation system (drip, sprinkler, and surface) or pumping system (diesel pumping and electric pumping) (MJ/m^3) represents the energy consumption of five different irrigation and pumping system to pump and deliver one cubic meter of water. **c** CO_2 emissions intensity is expressed as the ratio of the carbon dioxide emitted by irrigation to the irrigated area in a country ($100 \text{ kg CO}_2/\text{ha}$). **d** the CO_2 emissions intensity of each irrigation system or pumping system is expressed in the same way as

the energy intensity ($\text{g CO}_2/\text{m}^3$). Energy and CO_2 emissions intensity reflect the average level during 2000–2010 period. Mean values in the boxplot are shown with diamonds and median values are shown with midlines. The dashed lines in the figure are used to distinguish countries that are above or below the median energy and CO_2 emissions intensity per unit of water use.

(2) Figure 2

According to the Reviewer’s suggestion, in terms of color, we removed color gradients and used pastel colors (Fig. 2). In addition, we incorporating energy and CO_2 emissions from sustainable irrigation expansion (as suggested by another reviewer) in Figure 2 using a stacked bar graph. After the improvement, the bar graph is basically visible.

Fig. 2 Global energy consumption and CO_2 emissions from irrigation. Country-level energy consumption and CO_2 emissions are based on pixel sum statistics. **a** Energy consumption (PJ per

year) under 2000-2010 and sustainable irrigation expansion of 3 °C warmer climate. **b** CO₂ emissions (Mt CO₂ per year) from energy consumption under 2000-2010 and sustainable irrigation expansion of 3 °C warmer climate. We selected the top 20 countries with the highest energy consumption and CO₂ emissions. The upper right subgraphs represent a summary of energy consumption and CO₂ emissions by regions as well as globally.

We appreciate for Editors/Reviewers' warm work earnestly, and hope that the correction will meet with approval.

Once again, thank you very much for your comments and suggestions.

Reviewers' Comments:

Reviewer #2:

Remarks to the Author:

You have addressed all of my comments, but there are some new updates in COP28 and Global Stocktake report on renewable energy policy toward the net-zero target by 2050. I would request you to revisit them and make changes in facts and figures where necessary.

Reviewer #3:

Remarks to the Author:

I appreciate the effort the authors put into these revisions, including addressing over 30 substantive comments from the two reviewers and doing some new analyses that resulted in a completely updated set of results. This sincere response reflects well on everyone, good work!

I am satisfied with the revisions. The figures are substantially improved, as are the methodology that underlies them. The authors have done a good job compiling data for comparison. Furthermore, the improvements to the Discussion section are appreciated.

I look forward to sharing this paper with collaborators and students in the future, should the Editor choose to publish it.

Response to Reviewers

Reviewers' comments:

Reviewer # 2:

1. You have addressed all of my comments, but there are some new updates in COP28 and Global Stocktake report on renewable energy policy toward the net-zero target by 2050. I would request you to revisit them and make changes in facts and figures where necessary.

Revised manuscript: Caveats Section in main text Line 607-621

Response: Firstly, we sincerely appreciate your time spent reviewing the revised manuscript and your recognition of our response. Additionally, in response to the reviewer's suggestion, we conducted a search on the content regarding renewable energy policies towards achieving the net-zero target by 2050, as reported by COP28 and the Global Stocktake.

According to the report, policies on sustainable energy mainly include:

(1) Declaration to Triple Nuclear Energy ^[69]

On 02 December 2023, more than 20 countries from four continents launched the Declaration to Triple Nuclear Energy working together to advance a goal of tripling nuclear energy capacity globally by 2050. Endorsing countries include the United States, Armenia, Bulgaria, Canada, Croatia, Czech Republic, Finland, France, Ghana, Hungary, Jamaica, Japan, Republic of Korea, Moldova, Mongolia, Morocco, Netherlands, Poland, Romania, Slovakia, Slovenia, Sweden, Ukraine, United Arab Emirates, and United Kingdom.

(2) Global renewables and Energy Efficiency Pledge ^[69]

As of 11 December, With the endorsement of 130 national governments, the Pledge stipulates that signatories commit to work together to triple the world's installed renewable energy generation capacity to at least 11,000 GW by 2030 and to collectively double the global average annual rate of energy efficiency improvements from around two per cent to over four per cent every year until 2030.

“In light of the ambitious renewable energy policies reported at COP28 (28th Conference of the Parties to the United Nations Framework Convention on Climate Change) aimed at achieving net-zero emissions targets by 2050 ^[69], our findings may underestimate the potential for carbon emissions reduction from irrigation by 2050. Notably, major consumer of irrigation energy—India, China, Pakistan, and Iran—have not committed to tripling nuclear energy nor increasing renewable power generation capacity threefold by 2030. Consequently, while global carbon emissions

from the energy sector are projected to significantly decrease by 2050, reducing carbon emissions from irrigation will necessitate these countries to make greater commitments towards renewable energy adoption. Furthermore, the renewable energy policies outlined at COP28 do not delineate the proportion of low-carbon electricity expected by 2050 for individual countries or continents, hindering precise estimations of the potential reduction in irrigation carbon emissions. Our study offers a framework for evaluating the potential reduction in irrigation-related carbon emissions, enabling updates to our findings as new data becomes available.”

Therefore, according to the Reviewer’s suggestion, in our revised Caveats section, we added the discussion on COP28 report. The details are in italics above:

Finally, we appreciate for your warm work earnestly, and hope that the correction will meet with approval.

References:

[69] UNFCCC. *Summary of global climate action at COP28*. (UNFCCC: Dubai, the United Arab Emirates, 2023)

https://unfccc.int/sites/default/files/resource/Summary_GCA_COP28.pdf

Reviewer #3:

1. I appreciate the effort the authors put into these revisions, including addressing over 30 substantive comments from the two reviewers and doing some new analyses that resulted in a completely updated set of results. This sincere response reflects well on everyone, good work!

I am satisfied with the revisions. The figures are substantially improved, as are the methodology that underlies them. The authors have done a good job compiling data for comparison. Furthermore, the improvements to the Discussion section are appreciated.

I look forward to sharing this paper with collaborators and students in the future, should the Editor choose to publish it.

Response: Thank you very much for taking your valuable time to review this revised manuscript. And we appreciate your approval of our revised manuscript.